# Bacterial rhomboid proteases mediate quality control of orphan membrane proteins

Guangyu Liu[1], Stephen E Beaton[2], Adam G Grieve[1], Rhiannon Evans[2], Miranda Rogers[1], Kvido Strisovsky[3] (ID), Fraser A Armstrong[2], Matthew Freeman[1] (ID), Rachel M Exley[1,*] & Christoph M Tang[1,**] (ID)

## Abstract

Although multiprotein membrane complexes play crucial roles in bacterial physiology and virulence, the mechanisms governing their quality control remain incompletely understood. In particular, it is not known how unincorporated, orphan components of protein complexes are recognised and eliminated from membranes. Rhomboids, the most widespread and largest superfamily of intramembrane proteases, are known to play key roles in eukaryotes. In contrast, the function of prokaryotic rhomboids has remained enigmatic. Here, we show that the *Shigella sonnei* rhomboid proteases GlpG and the newly identified Rhom7 are involved in membrane protein quality control by specifically targeting components of respiratory complexes, with the metastable transmembrane domains (TMDs) of rhomboid substrates protected when they are incorporated into a functional complex. Initial cleavage by GlpG or Rhom7 allows subsequent degradation of the orphan substrate. Given the occurrence of this strategy in an evolutionary ancient organism and the presence of rhomboids in all domains of life, it is likely that this form of quality control also mediates critical events in eukaryotes and protects cells from the damaging effects of orphan proteins.

**Keywords** intramembrane proteolysis; membrane protein complexes; quality control; rhomboid; *Shigella*
**Subject Categories** Microbiology, Virology & Host Pathogen Interaction; Post-translational Modifications & Proteolysis; Translation & Protein Quality
**The EMBO Journal (2020) 39: e102922**

See also J Begalla *et al* (May 2020) and JD Knopf & MK Lemberg (May 2020)

## Introduction

Multiprotein membrane complexes in bacteria mediate fundamental processes such as respiration, secretion of virulence factors and nutrient acquisition (Unden *et al*, 2014; Costa *et al*, 2015; Sheldon

*et al*, 2016). Aberrant assembly or disassembly of complexes results in orphan proteins, which usually require prompt degradation to maintain cellular proteostasis (Harper & Bennett, 2016; Juszkiewicz & Hegde, 2018). However, little is known how unincorporated, orphan components of bacterial protein complexes are sensed and eliminated from membranes.

Rhomboids are the largest family of intramembrane proteases (IMPs) and are found in all kingdoms of life. These enzymes rapidly scan membranes for their substrates (Kreutzberger *et al*, 2019) and have their active sites embedded in the lipid bilayer where they cleave their substrates in or adjacent to transmembrane domains (TMDs). A wide range of functions has been ascribed to eukaryotic rhomboids including growth factor signalling (Urban *et al*, 2001), lipid metabolism (Saita *et al*, 2018), energy production (Spinazzi *et al*, 2019), chloroplast development (Thompson *et al*, 2012), apoptosis regulation (Saita *et al*, 2017), endoplasmic reticulum (ER) protein trafficking (Fleig *et al*, 2012) and surface antigen shedding in the apicomplexan parasites (Shen *et al*, 2014).

Most insights into the catalytic mechanism of rhomboids have been gained from studies of the *Escherichia coli* rhomboid, GlpG (Wang *et al*, 2006; Wu *et al*, 2006; Ben-Shem *et al*, 2007; Xue & Ha, 2012; Zoll *et al*, 2014; Cho *et al*, 2016). The serine residue of the catalytic dyad, $Ser^{201}/His^{254}$, is embedded in rhomboids approximately 10 Å below the surface of the membrane (Wu *et al*, 2006; Ben-Shem *et al*, 2007). Current models indicate that rhomboid-mediated proteolysis is a rate-driven process, with the affinity of the enzyme for its substrate not playing an important role (Dickey *et al*, 2013; Cho *et al*, 2016). An initial "interrogation complex" is formed once the TMD of a substrate engages GlpG with an accessible catalytic site (Strisovsky *et al*, 2009; Dickey *et al*, 2013). Subsequent transition to a "scission complex" necessitates unwinding of the TMD of the substrate, driven by helix-destabilising residues such as prolines (Strisovsky *et al*, 2009; Moin & Urban, 2012; Cho *et al*, 2016). Additionally, the nature of residues at certain positions in the TMD of substrates has a major impact on cleavage (Strisovsky *et al*, 2009). In particular, the P1 residue, defined as the newly formed C-terminal residue upon

1 Sir William Dunn School of Pathology, University of Oxford, Oxford, UK
2 Inorganic Chemistry Laboratory, University of Oxford, Oxford, UK
3 Institute of Organic Chemistry and Biochemistry, Academy of Sciences of the Czech Republic, Praha 6, Czech Republic
*Corresponding author. Tel: +44 1865 275500; Email: rachel.exley02@path.ox.ac.uk
**Corresponding author. Tel: +44 1865 275500; Email: christoph.tang@path.ox.ac.uk
[The copyright line of this article was changed on 18 May 2020 after original online publication]

cleavage, must be a small aliphatic residue, with this requirement applying to rhomboids from divergent evolutionary backgrounds (Strisovsky *et al*, 2009; Riestra *et al*, 2015; Saita *et al*, 2017). However, further understanding of the mechanism of rhomboid proteolysis is hampered by the lack of knowledge of the cognate substrate(s) of GlpG.

Despite the near-universal presence of rhomboids in bacteria (Koonin *et al*, 2003), remarkably little is known about the role of these enzymes in prokaryotes. To date, only a single substrate of a bacterial rhomboid is known; AarA is a rhomboid in *Providencia stuartii* which cleaves TatA (Stevenson *et al*, 2007), an essential component of the twin-arginine translocation (Tat) system (Palmer & Berks, 2012). TatA processing by AarA is critical for the function of the Tat system, which mediates quorum sensing in *P. stuartii* (Stevenson *et al*, 2007). Despite current knowledge of GlpG structure and activity, the only phenotypes identified for *E. coli glpG* mutants are an enhanced resistance to cefotaxime (Clemmer *et al*, 2006) and reduced intestinal colonisation in a murine model (Russell *et al*, 2017). However, the molecular mechanisms underlying these phenotypes are unknown.

Here, we identified and characterised two rhomboids, GlpG and Rhom7, in *Shigella sonnei*, a close relative of *E. coli* that causes bacillary dysentery (Kotloff *et al*, 2017). Similar to *E. coli*, *S. sonnei* colonises the anaerobic environment of the large intestine, where it can invade the epithelial surface by virtue of its type three secretion system (T3SS) (Marteyn *et al*, 2010). GlpG in *S. sonnei* has 99% amino acid identity with the prototypical rhomboid of *E. coli*, while Rhom7 is predicted to possess seven TMDs. A screen for their substrates identified components of three membrane respiratory complexes: HybA and FdoH for GlpG, and HybA and FdnH for Rhom7 (Abaibou *et al*, 1995; Jormakka *et al*, 2002; Pinske *et al*, 2015). We found that rhomboids exhibit exquisite selectivity by cleaving orphan substrates that are dissociated from their cognate complex, while leaving functional substrates intact. Our findings reveal that rhomboids contribute to the quality control of multiprotein membrane complexes and membrane proteostasis, with rhomboid-mediated proteolysis serving as the critical licensing step that allows downstream proteolytic degradation of orphan substrates, so preventing their aggregation.

# Results

## *Shigella sonnei* possesses two active rhomboids, GlpG and Rhom7

To identify rhomboid proteases in *S. sonnei*, BLASTp searches were performed using sequences of *E. coli* GlpG (accession no. YP_026220.1) and *P. stuartii* AarA (accession no. AAA61597.1). The *S. sonnei* proteome (taxid: 300269) was used for initial analysis. BLASTp revealed two homologues in *S. sonnei*: SSON_3661 and SSON_0610. SSON_3661 differs by a single amino acid (a.a.) from *E. coli* GlpG (Ala[130] in *S. sonnei* versus Thr[130] in *E. coli*) and so was designated GlpG (Fig EV1). SSON_0610 shares 30% amino acid identity with *P. stuartii* AarA and harbours two potential catalytic residues, Ser[133] and His[187] (Fig EV1). Topology prediction by Phobius (Kall *et al*, 2007) suggests that SSON_0610 possesses seven TMDs, as opposed to GlpG which contains six TMDs (Wang *et al*, 2006). Therefore, SSON_0610 was designated Rhom7.

To establish whether *Shigella* GlpG and Rhom7 are active IMPs, we examined their ability to cleave an artificial substrate (AS) consisting of an N-terminal maltose-binding protein (MBP), a triple-FLAG tag (3xFLAG), the TMD of *Providencia stuartii* TatA (a.a. 1-50) and a thioredoxin domain (Trx) (Fig 1A) (Strisovsky *et al*, 2009). The AS was introduced on a plasmid into *S. sonnei* lacking chromosomal copies of *glpG* and *rhom7* (*S. sonnei* Δ*glpG*Δ*rhom7*) with these genes expressed from plasmids instead; controls included strains expressing inactive GlpG (GlpG[S201A]) (Dickey *et al*, 2013) or with an empty vector. Cleavage of the AS was detected by Western blot analysis with anti-FLAG mAbs (Strisovsky *et al*, 2009) (Fig 1B and C). Both GlpG and Rhom7 cleaved the TMD of TatA within the AS, proving that both are active rhomboid proteases. GlpG[S201A] and Rhom7 with alanine substitution of either of the predicted catalytic residues (*i.e.* Rhom7[S133A] or Rhom7[H187A]) failed to cleave the AS (Fig 1B and C). Furthermore, we examined the effect of removing the 7th TMD and the C-terminal domain (Rhom7[ΔTM7]), or the C-terminal domain alone (Rhom7[ΔCTD]) of Rhom7. Results demonstrate that neither of these features is required for Rhom7 activity against the AS (Fig 1D).

## A bioinformatic screen identifies putative substrates for GlpG and Rhom7

To define the role of these rhomboids, we subjected *S. sonnei* Δ*glpG*Δ*rhom7* to multiple phenotypic assays including growth in complete/minimal media under aerobic or anaerobic conditions and performed Biolog MicroArrays comparing its behaviour with the wild-type strain. Deletion of the rhomboids had no effect in these assays (Fig EV2). As *glpG* is in an operon with *glpE*, which encodes a sulphur donor for the antioxidant thioredoxin 1 (Ray *et al*, 2000), we also tested whether GlpG and/or Rhom7 are involved in survival during oxidative stress. However, there was no significant difference in the recovery of wild-type *S. sonnei* and *S. sonnei* Δ*glpG*Δ*rhom7* after exposure to hydrogen peroxide or paraquat (Fig EV3). Furthermore, loss of GlpG and Rhom7 does not alter secretion through the type three secretion system (Fig EV4) which is essential for *Shigella* virulence (Schroeder & Hilbi, 2008). Taken together, we found GlpG and Rhom7 have no detectable impact on *Shigella* under a variety of conditions.

Therefore, we searched for the substrates of GlpG and Rhom7. Rhomboids have a propensity to cleave single-pass membrane proteins with a periplasmic N-terminus and a cytosolic C-terminus, *i.e.* membrane proteins with type I or type III topology (Urban & Freeman, 2003). To identify potential substrates, we looked for candidates based on their likely location and membrane topology (Fig 2A). The predicted proteome of *S. sonnei* SS046 was interrogated, and graphical representations of the topology of all proteins (*n* = 4,911) were generated using the TMHMM Server (Krogh *et al*, 2001). We manually selected type I and type III proteins, while excluding proteins with ambiguous assignments. This yielded 16 initial potential rhomboid substrates. To identify further candidates for subsequent analysis, the predicted TMDs of these 16 proteins were then used to identify further type I and III proteins using the HHpred server (Soding *et al*, 2005). Additional candidates identified by HHpred analysis were further analysed with Phobius 1.01 and TOPCONS 2.0 to detect signal peptides and to refine topology predictions (Kall *et al*, 2007; Tsirigos *et al*, 2015). Homology

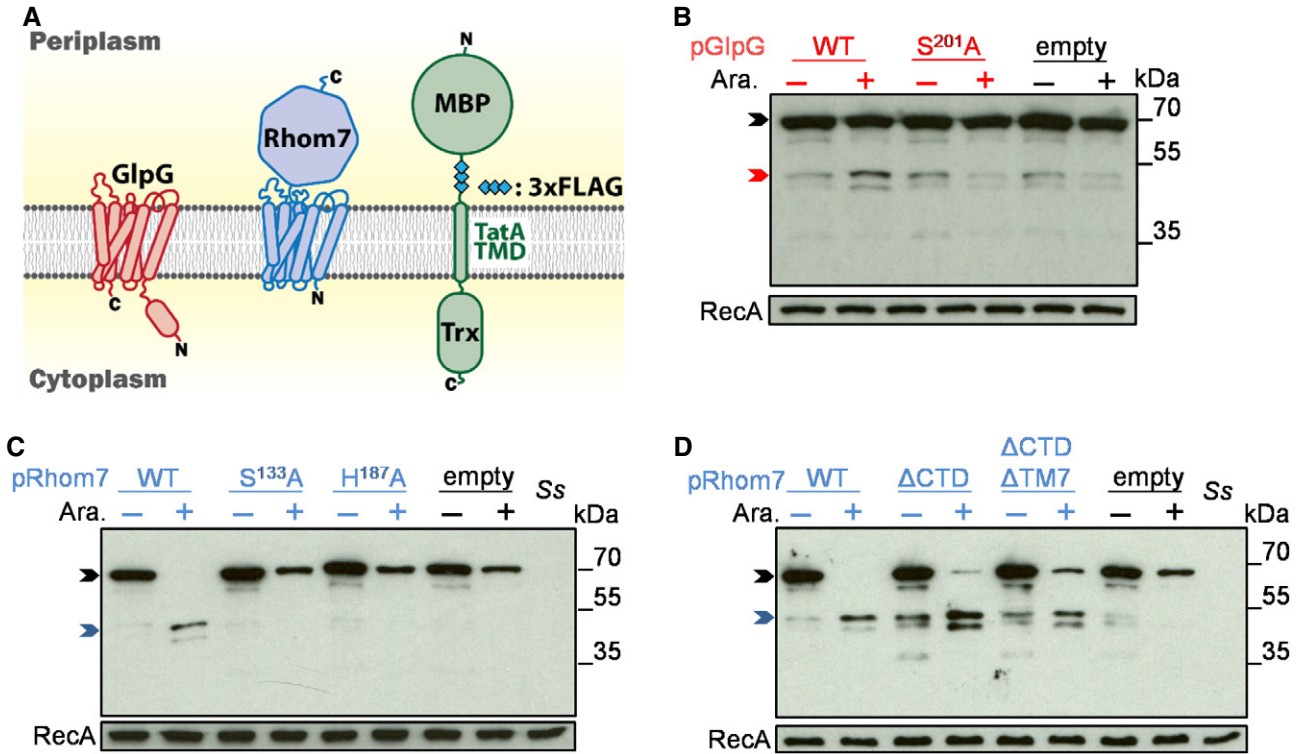

**Figure 1. GlpG and Rhom7 are active rhomboids.**

A Topology of GlpG, Rhom7 and the artificial substrate with a maltose-binding protein (MBP) domain, triple-FLAG tag (3xFLAG), the TMD of *P. stuartii* TatA and a thioredoxin domain (Trx).

B Western blot analysis (probing with an anti-FLAG mAb) to detect cleavage of the artificial substrate by wild-type (WT)/inactive (S201A) GlpG encoded on pBAD33 with/without arabinose (Ara.).

C Western blot analysis to detect cleavage of the artificial substrate by wild-type (WT)/modified (S133A or H187A) Rhom7 encoded on pBAD33 with/without arabinose (Ara.).

D Activity of Rhom7 with/without its 7th TMD and/or C-terminal domain.

Data information: In (B-D), rhomboid substrates that are uncleaved, cleaved by GlpG or cleaved by Rhom7 are marked by black, red and blue arrows, respectively. Controls, empty pBAD33 (empty) and wild-type *Shigella sonnei* (*Ss*). RecA, loading control.

Source data are available online for this figure.

searches with HHpred were performed reiteratively until no further candidates with type I or type III topology were identified, resulting in 44 potential rhomboid substrates.

To examine whether the TMDs of these proteins are cleaved by GlpG and/or Rhom7, the TatA TMD in the AS was replaced with the predicted TMD plus 16 residues at the periplasmic aspect of each candidate (Fig 2B and Table EV1). Western blot analysis was used to assess cleavage of the candidates' TMDs in *S. sonnei* lacking *glpG* and *rhom7* with plasmid-encoded active (wild-type) or inactive versions of the proteases (GlpG$^{S201A}$ and Rhom7$^{S133A}$) (Fig 2B and Appendix Fig S1). A positive hit was defined when more product and less full length substrate were observed in the presence of an active rhomboid compared to an inactive enzyme. TMDs from six candidates were reproducibly cleaved by GlpG and/or Rhom7 (Fig 2C–E). Plasmid-encoded GlpG cleaved the TMDs from five candidates: HybA and HybO, subunits of the hydrogenase-2 complex (Hyd-2) (Pinske *et al*, 2015); FdoH, a subunit of the formate dehydrogenase O complex (Abaibou *et al*, 1995); YqjD, a ribosome-associated protein (Yoshida *et al*, 2012);

and YtjC, a putative phosphatase (Yip & Matsumura, 2013). Plasmid-encoded Rhom7 also cleaved the TMDs from HybO and HybA, as well as FdnH, a component of formate dehydrogenase N (Jormakka *et al*, 2002).

**HybA is a physiological substrate of GlpG**

Our screen for TMDs cleaved by over-expressed GlpG or Rhom7 identified two proteins belonging to the Hyd-2 complex, HybA and HybO (Pinske *et al*, 2015). Therefore, we next analysed whether full-length HybO and HybA are genuine rhomboid substrates. We constructed strains in which the chromosomal, native genes encoding HybO or HybA were tagged with a sfCherry-3xFLAG tag to allow detection of their expression and cleavage by Western blotting (Fig 3A); GlpG and Rhom7 were expressed from their native locus or from a plasmid. As Hyd-2 is only expressed in the absence of oxygen (Richard *et al*, 1999), we examined cleavage under anaerobic conditions. When analysing HybO cleavage, there was a band of unknown origin with the same molecular mass as predicted for

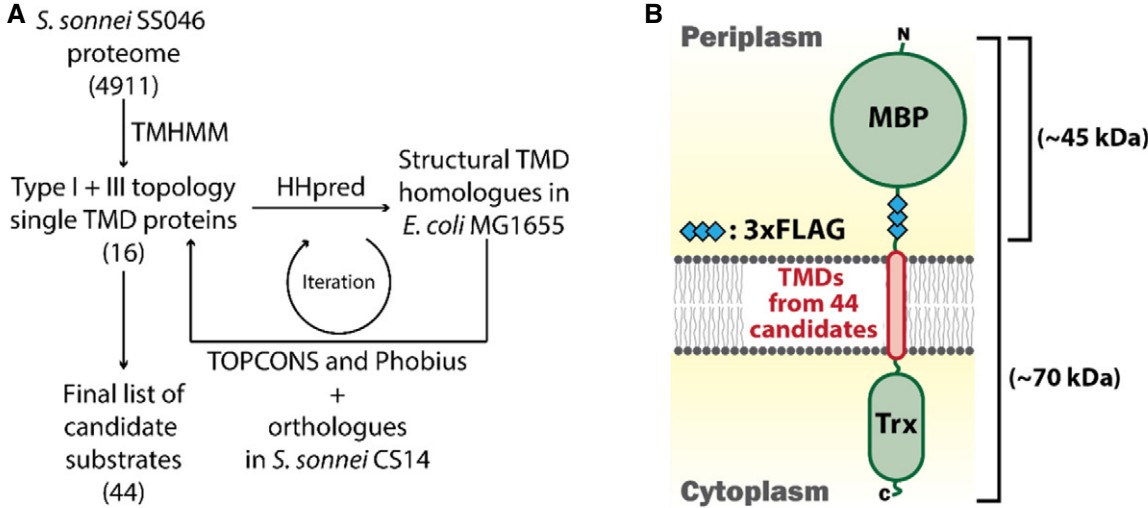

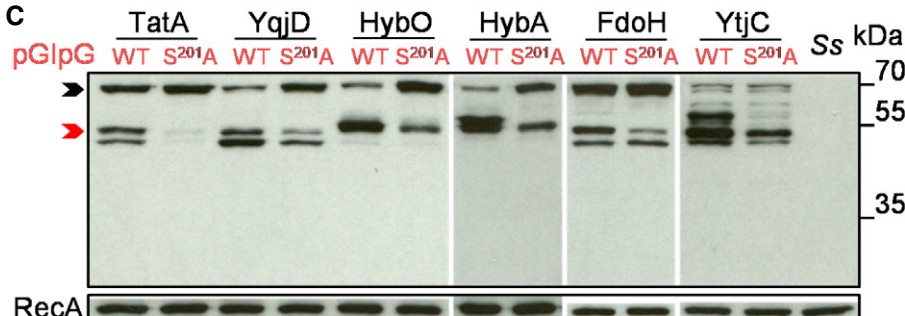

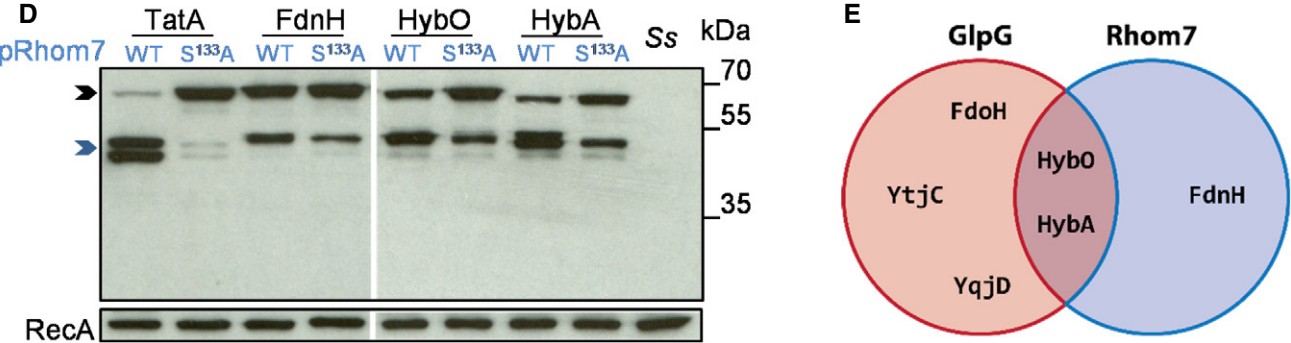

**Figure 2. Identification of potential GlpG and Rhom7 substrates.**

A Workflow of the bioinformatic identification of candidate rhomboid substrates.

B A library of artificial substrates harbouring TMDs from 44 candidate rhomboid substrates.

C Western blot analysis of candidate substrates reproducibly cleaved by GlpG from the screen.

D Western blot analysis of candidate substrates reproducibly cleaved by Rhom7 from the screen.

E Venn diagram summarising putative rhomboid substrates identified by the screen.

Data information: In (C and D), rhomboid substrates that are uncleaved, cleaved by GlpG or cleaved by Rhom7 are marked by black, red and blue arrows, respectively. Controls, inactive enzymes and wild-type *Shigella sonnei* (*Ss*). RecA, loading control.

Source data are available online for this figure.

cleaved HybO even in the absence of GlpG and Rhom7 (Fig 3B). Consequently, we were unable to assess whether either rhomboid cleaves full-length HybO. However, HybA was cleaved by both chromosomally and plasmid-encoded GlpG, as well as by plasmid-encoded Rhom7 (Fig 3C). We mapped the GlpG cleavage site in the TMD of HybA by N-terminal sequencing and identified Gly[296] as the

P1 residue (Fig EV5), *i.e.* the C-terminal residue generated upon cleavage, which is consistent with preferred P1 residues in bacterial rhomboid substrates (Strisovsky *et al*, 2009). Characteristically of rhomboid substrates, substitution of the P1 residue with a bulky amino acid (HybA^G296F) rendered HybA resistant to cleavage by chromosomally or plasmid-expressed GlpG (Fig 3D). Another general feature of rhomboid substrates is the presence of a helix-destabilising residue in the TMD (Moin & Urban, 2012). The HybA TMD contains a potentially destabilising proline, and indeed, HybA^P300A was not cleaved by GlpG or Rhom7 (Fig 3E). Taken together, our results confirm HybA as the first substrate of GlpG and Rhom7.

## GlpG specifically targets orphan HybA and does not affect Hyd-2 activity

Despite the visible cleavage of HybA, we noticed that > 80% of endogenous HybA remained uncleaved even when GlpG or Rhom7 was expressed from multicopy plasmids (Fig 3C). Considering that the substrate TMD must be accessible to the active site of the rhomboid in the lipid bilayer, we hypothesised that HybA might be protected when it is part of the Hyd-2 complex, but in a protease-sensitive state when it is an isolated, orphan protein. To test this hypothesis, we performed bacterial two-hybrid analysis and demonstrated that HybA interacts directly with HybB (Fig 4A) consistent

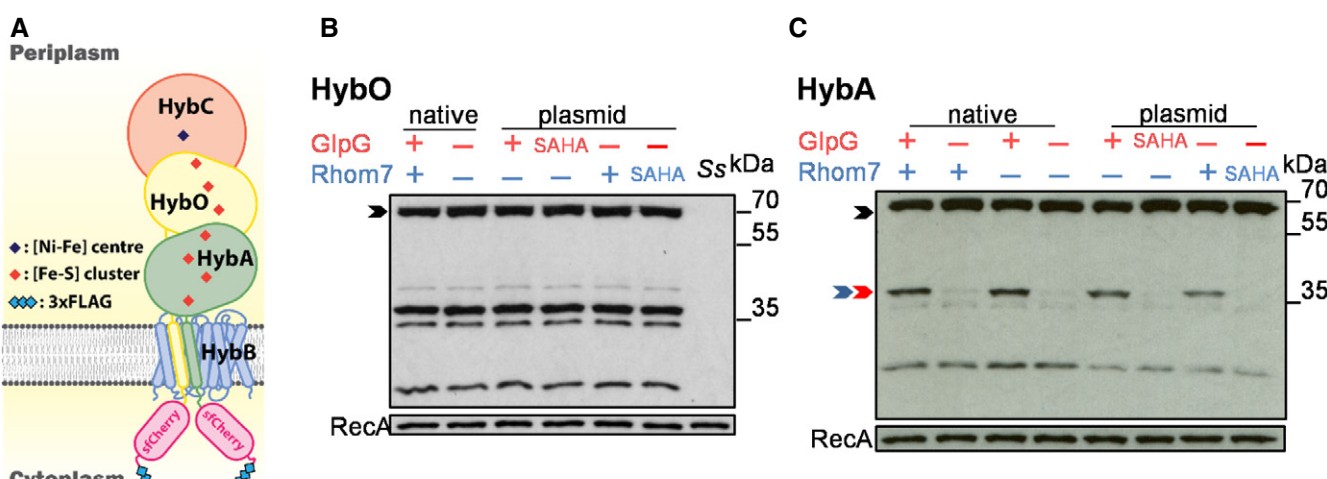

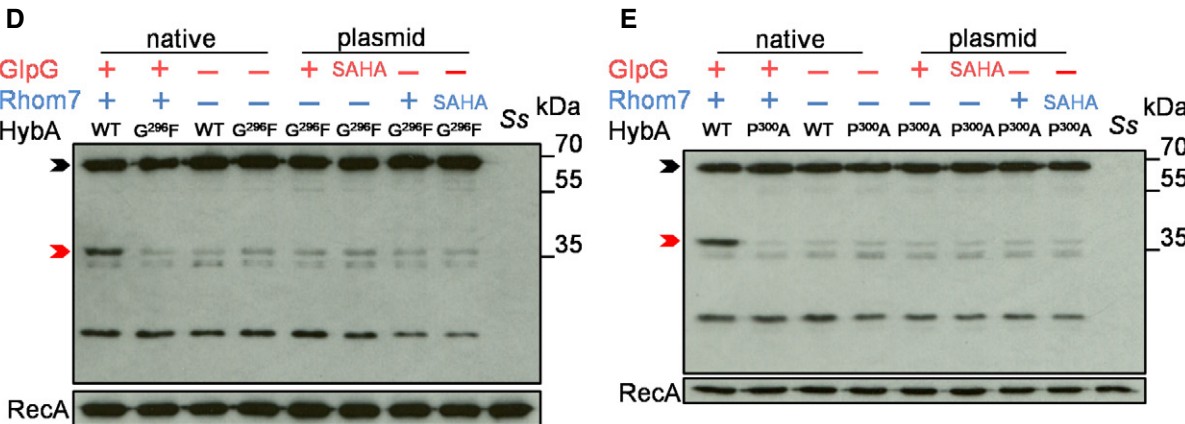

**Figure 3. HybA is a physiologic substrate of GlpG.**

A   Schematic of HybA and HybO fusions.
B   Western blot analysis (probing with an anti-FLAG mAb) to detect cleavage of HybO with (+)/without (−) chromosomal (native) or pBAD33-encoded rhomboids (plasmid).
C   Western blot analysis to detect cleavage of HybA with (+)/without (−) chromosomal (native) or pBAD33-encoded rhomboids (plasmid).
D   Western blot analysis to detect cleavage of HybA^G296F by endogenous or pBAD33-encoded rhomboid.
E   Western blot analysis to detect cleavage of HybA^P300A by endogenous or pBAD33-encoded rhomboid.

Data information: In (B-E), rhomboid substrates that are uncleaved, cleaved by GlpG or cleaved by Rhom7 are marked by black, red and blue arrows, respectively. Wild-type *Shigella sonnei, Ss.* Wild-type (WT)/inactive (SAHA: alanine substitution of the catalytic serine and histidine residues) enzymes were pBAD33-encoded (plasmid) in *S. sonnei ΔglpGΔrhom7*. RecA, loading control.
Source data are available online for this figure.

with previous biochemical data (Pinske *et al*, 2015). Next, we manipulated the stoichiometry of the Hyd-2 complex by either over-expressing HybA or removing HybB. Strikingly, deletion of *hybB* led to a ≈five-fold increase in the proportion of cleaved to uncleaved HybA (Fig 4B and C). Consistent with this, over-expression of HybA in the presence of HybB resulted in a threefold increase in cleaved to

uncleaved HybA (Fig 4B and Appendix Fig S2A). Furthermore, when HybA is not replenished by *de novo* protein synthesis (by adding chloramphenicol to cells, Fig 4D and E), it is apparent that GlpG cleaves a large proportion of HybA when it is an orphan protein—*i.e.* in the absence of HybB (Fig 4D and Appendix Fig S2B). In contrast, HybA remains largely uncleaved in the presence of HybB (Fig 4E).

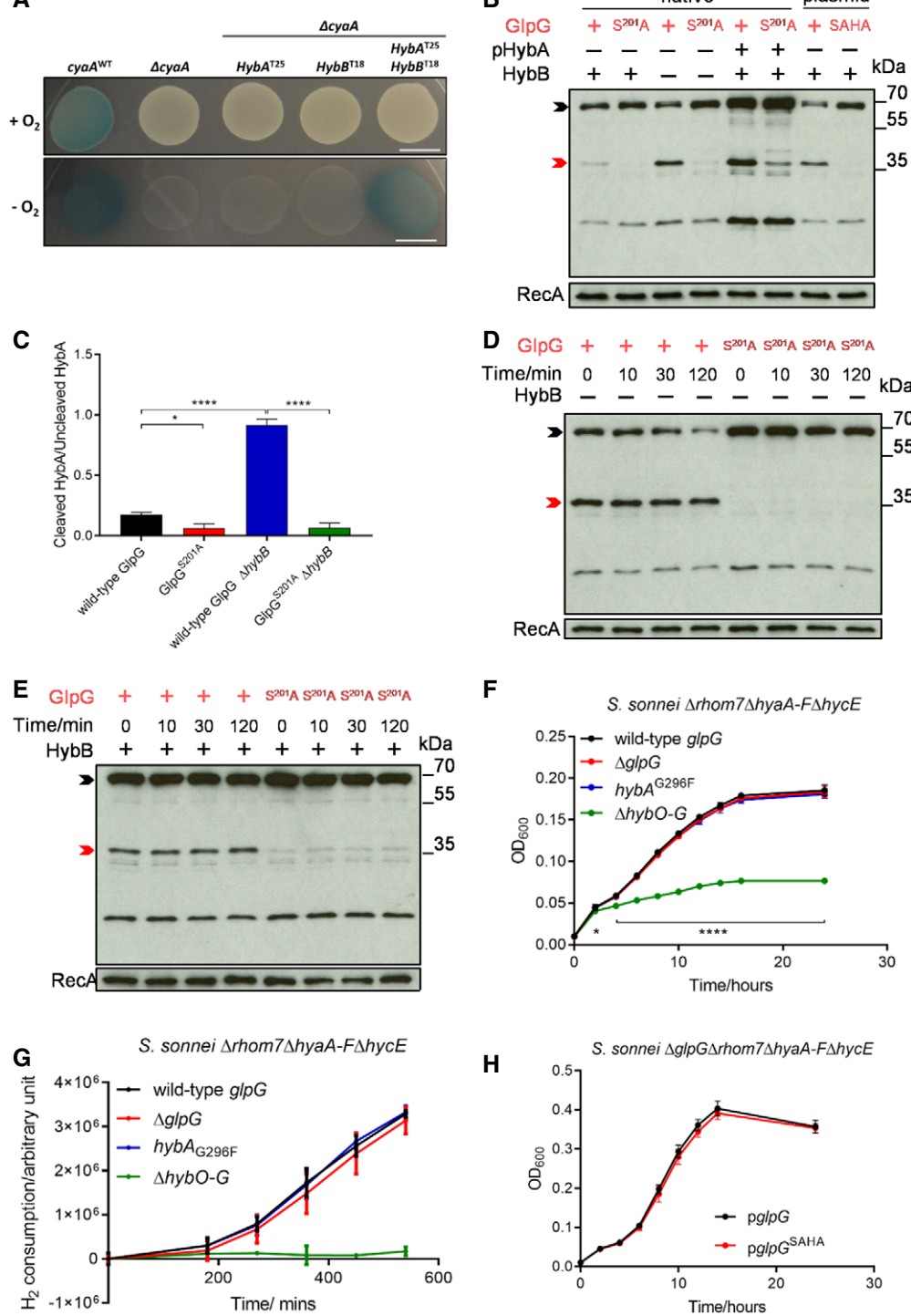

**Figure 4.**

◄

**Figure 4.  GlpG specifically targets orphan HybA and does not influence hydrogenase-2 activity.**

A    Bacterial two-hybrid analysis with HybA and/or HybB fused chromosomally with the T25 and T18 domains of *B. pertussis* CyaA, respectively, in the absence of endogenous CyaA. Bacteria were grown on LB agar containing 20 µg/ml X-gal at 37°C for 14 h in the presence/absence of $O_2$. Scale bar, 1 cm.

B    Western blot analysis to detect HybA cleavage in *S. sonnei* Δ*rhom7* by wild-type (+) or inactive ($S^{201}A$) GlpG expressed chromosomally (native) or from pUC19 (plasmid) with (+)/without (−) HybB. HybA was expressed from its native locus or a plasmid (pHybA).

C    Quantification of the ratio of cleaved/uncleaved HybA in strains with (+) or without (−) HybB. Mean ± S.D. of three experiments. *$P < 0.05$; ****$P < 0.0001$ (one-way ANOVA).

D    Western blot analysis to detect GlpG-mediated cleavage of native C-terminally sfCherry-3xFLAG-tagged HybA at indicated times after blocking protein translation by the addition of chloramphenicol at $T_0$ in the absence of HybB (−).

E    Western blot analysis to detect GlpG-mediated cleavage of tagged HybA at times after blocking protein translation by the addition of chloramphenicol at $T_0$ in the presence of HybB (+).

F    Growth of bacteria lacking hydrogenases (Δ*hyaA-F* Δ*hycE* +/− Δ*hybO-G*) or GlpG (Δ*glpG*), or expressing uncleavable HybA (*hybA*$^{G296F}$) in 5% $H_2$

G    $H_2$ consumption by *S. sonnei*. Bacteria were grown aerobically in LB overnight and then diluted into M9 minimal media supplemented with 0.5% fumarate, 12.5 µg/ml nicotinic acid and 0.2% casamino acids in a sealed glass chromatography vial. The headspace was purged with 10% $H_2$/90% argon, and cultures were incubated at 37 °C with shaking at 180 rpm for 9 h. $H_2$ in the headspace was sampled and measured by gas chromatography.

H    *Shigella sonnei* growth in 10% $H_2$ with plasmid-expressed wild-type or non-functional (pglpG$^{SAHA}$) GlpG.

Data information: *P*-values were calculated by two-way ANOVA (F, H). Bars represent the mean ± SD *n* = 3 (F-H); *$P < 0.05$; ****$P < 0.001$. In (B, D, and E), HybA that is uncleaved or cleaved by GlpG is marked by black and red arrows, respectively. RecA, loading control.

Source data are available online for this figure.

If GlpG selectively cleaves HybA when it is an orphan molecule, GlpG would be predicted to not affect Hyd-2 activity even though HybA is an essential component of the Hyd-2 complex (Pinske *et al*, 2015). To test this, we assessed whether GlpG affects anaerobic growth with $H_2$ as the main electron donor (Pinske *et al*, 2015) of strains with Hyd-2 as the only hydrogenase (in *S. sonnei* Δ*hyaA-F*Δ*hycE*), and with GlpG as the only rhomboid (by deleting *rhom7*). Under these conditions, the growth of *S. sonnei* is Hyd-2-dependent as illustrated by the severe growth defect of the strain lacking Hyd-2 (Δ*hybO-G*) (Fig 4F). Of note, deletion of *glpG* or expression of uncleavable HybA$^{G296F}$ had no detectable impact on growth (Fig 4F). Furthermore, we measured bacterial $H_2$ consumption, a more sensitive and direct readout of Hyd-2 activity (Pinske *et al*, 2015) using the same *S. sonnei* strains during $H_2$-dependent growth under anaerobic conditions. We observed no significant difference in the $H_2$ consumption of strains lacking GlpG or expressing non-cleavable HybA (Fig 4G); under these conditions, wild-type HybA was still cleaved by GlpG, and there was no difference in the survival of strains (Appendix Fig S3). Remarkably, even expression of GlpG from a multicopy plasmid had no impact on Hyd-2 function (Fig 4H), highlighting that GlpG has specifically evolved to target non-functional HybA.

## GlpG and Rhom7 also target orphan components of formate dehydrogenases

Our data illustrate that GlpG is involved in quality control, by selectively targeting a member of a multi-subunit membrane complex when it is an orphan protein (Juszkiewicz & Hegde, 2018). Interestingly, our initial rhomboid substrate screen also identified other components of related multiprotein complexes (Fig 2C–E), *i.e.* FdoH, part of formate dehydrogenase O with FdoG and FdoI (Abaibou *et al*, 1995), and FdnH, part of formate dehydrogenase N with FdnG and FdnI (Jormakka *et al*, 2002). This led us to test whether the targeting of orphan components of multiprotein complexes by rhomboids is a general mechanism of quality control. We constructed strains in which FdoH or FdnH were tagged with the sfCherry-3xFLAG tag (Fig 5A) and assessed whether rhomboids cleave FdoH and FdnH when the stoichiometry of their complexes is perturbed. Notably,

GlpG cleavage of FdoH only became evident in the absence of FdoI (Fig 5B), while plasmid-expressed Rhom7 processes FdnH in bacteria lacking its partner FdnI (Fig 5C). Similar to HybA, the TMDs of FdnH and FdoH have helix-destabilising residues (Pro$^{259}$ in both), which are conserved in HybA/FdnH/FdoH orthologues in phylogenetically distant bacterial species as well as in *P. stuartii* TatA (Fig 5D). Importantly, mutating these hallmarks of rhomboid substrates (Moin & Urban, 2012) (HybA$^{P300A}$, FdnH$^{P259A}$ and FdoH$^{P259A}$) renders them resistant to cleavage by GlpG and Rhom7 even when they are orphan proteins (Fig 5E–G). In conclusion, our data demonstrate that *Shigella* rhomboids mediate quality control of orphan components from multiple multiprotein respiratory complexes and that cleavage is dependent on an evolutionarily conserved proline residue embedded in TMD of the substrate.

## Rhomboid cleavage licenses further degradation of substrates and prevents the formation of membrane aggregates of orphan HybA

To examine the fate of substrates after initial rhomboid cleavage, we inserted an N-terminal V5 epitope tag into natively expressed wild-type HybA or HybA$^{P300A}$ to enable detection of the periplasmic domain. Orphan HybA was cleaved by wild-type GlpG, then subjected to further degradation as evidenced by a reduction in the amount of uncleaved HybA and the appearance of bands of lower molecular mass than the GlpG-mediated cleavage product (Fig 6A). In contrast, there was no evidence of either initial cleavage or further processing of HybA$^{P300A}$. Further degradation of GlpG-cleaved HybA was particularly evident when *de novo* protein translation was blocked by the addition of chloramphenicol (Fig 6B). We employed the same strategy to follow FdoH and FdnH following cleavage by GlpG and Rhom7, respectively. Interestingly, there was no evidence that cleaved, orphan FdoH or FdnH were subject to further degradation when bacteria were grown aerobically or anaerobically in LB supplemented with nitrate (Fig 6C and D, respectively). However, there was detectable processing of both these substrates when bacteria were grown anaerobically in LB supplemented with glycerol and fumarate and then subjected to brief copper stress, which is known to perturb the stability of Fe-S

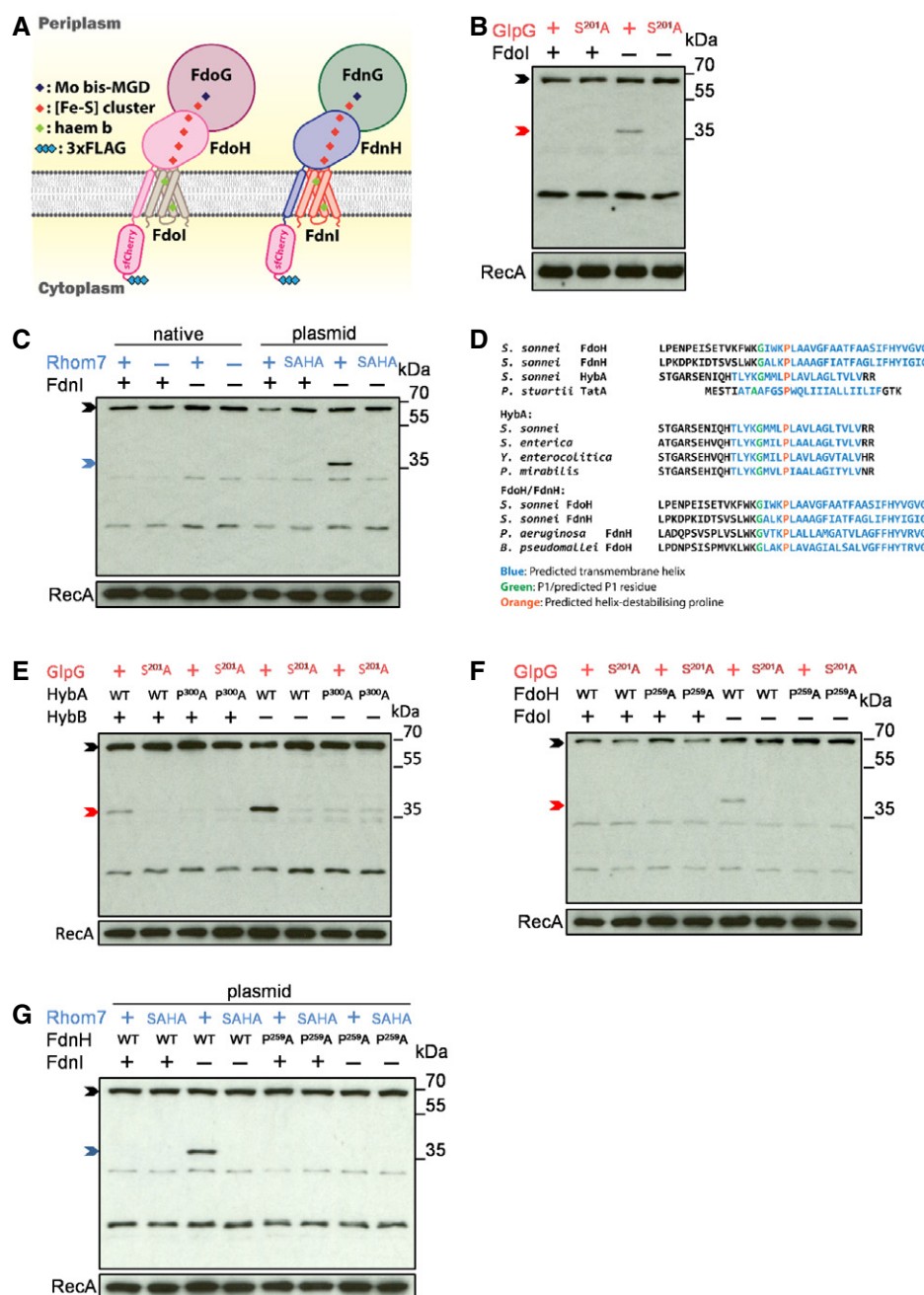

**Figure 5. GlpG and Rhom7 also target orphan components of formate dehydrogenases.**

A  Schematic of FdoH and FdnH fusions.

B  Western blot analysis (probing with an anti-FLAG mAb) to detect cleavage of FdoH cleavage in *S. sonnei* Δ*rhom7* with (+)/without (−) FdoI.

C  Western blot analysis to detect cleavage of FdnH in *S. sonnei* Δ*glpG* with (+) or without (−) chromosomal Rhom7 (native) or wild-type (+)/inactive (SAHA) Rhom7 expressed from pBAD33 in *S. sonnei* Δ*glpG*Δ*rhom7* with (+) or without (−) FdnI.

D  Alignments of HybA, FdoH and FdnH in *S. sonnei* with *Providencia stuartii* TatA and with their homologues from phylogenetically diverse organisms (*S. sonnei, Salmonella enterica, Yersinia enterocolitica, Proteus mirabilis, Pseudomonas aeruginosa* and *Burkholderia pseudomallei*) highlighting conserved glycine (green) and proline (orange) residues.

E  Western blot analysis to detect cleavage of wild-type (WT) or uncleavable (P300A) HybA by active (+) or inactive (S201A) chromosomal GlpG in the presence (+) or absence (−) of HybB.

F  Western blot analysis to detect cleavage of wild-type (WT) or modified (P259A) FdoH by active (+) or inactive (S201A) chromosomal GlpG in the presence (+) or absence (−) of FdoI.

G  Western blot analysis to detect cleavage of wild-type (WT) or modified (P259A) FdnH with active (+) or inactive (SAHA) Rhom7 expressed from pBAD33 with (+) or without (−) FdnI.

Data information: In (B, C, E–G), rhomboid substrates that are uncleaved, cleaved by GlpG or cleaved by Rhom7 are marked by black, red and blue arrows, respectively. RecA, loading control.

Source data are available online for this figure.

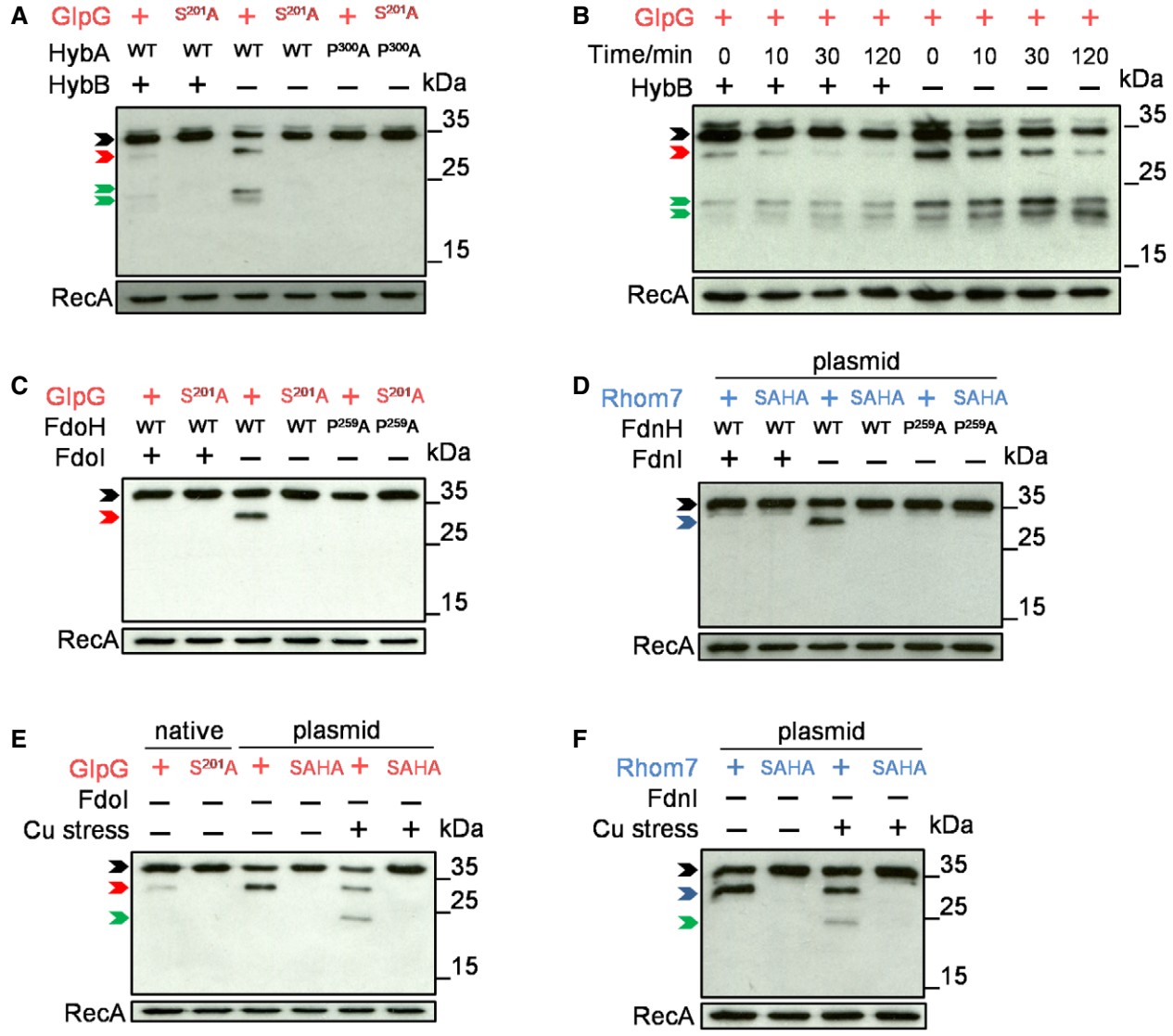

**Figure 6. Rhomboid cleavage licenses further degradation of substrates.**

A   Western blot analysis (probing with an anti-V5 mAb) to detect degradation of N-terminally V5-tagged wild-type (WT) or modified (P[300]A) HybA in *S. sonnei* Δ*rhom7* chromosomally expressing wild-type (+) or inactive (S[201]A) GlpG with (+) or without (−) HybB.

B   Degradation of V5-tagged HybA at times after blocking protein translation at $T_0$ in the presence (+) or absence (−) of HybB.

C   Western blot analysis of N-terminally V5-tagged wild-type (WT) or modified (P[259]A) FdoH in *S. sonnei* Δ*rhom7* chromosomally expressing wild-type (+) or inactive (S[201]A) GlpG with (+)/without (−) FdoI.

D   Western blot analysis of N-terminally V5-tagged wild-type (WT) or modified (P[259]A) FdnH in *S. sonnei* Δ*rhom7* with wild-type (+) or inactive (SAHA) Rhom7 expressed from pBAD33 with (+)/without (−) FdnI.

E, F   Degradation of N-terminally V5-tagged wild-type (WT) or modified (P[259]A) FdoH (E) or FdnH (F) in *S. sonnei* Δ*rhom7* with wild-type (+) or inactive (S[201]A, SAHA) GlpG expressed chromosomally (native) or from pUC19 (plasmid) without FdoI or FdnI (−), respectively, +/− exposure to 400 μM $CuCl_2$ for 30 min.

Data information: Rhomboid substrates that are uncleaved, cleaved by GlpG or cleaved by Rhom7 are marked by black, red and blue arrows, respectively. Degradation products post-rhomboid cleavage are marked by green arrows. RecA, loading control.
Source data are available online for this figure.

clusters (Macomber & Imlay, 2009). Therefore, initial rhomboid-mediated cleavage of orphan components of respiratory complexes licenses their further processing.

Given the selective activity of GlpG against orphan HybA, we reasoned that a potential role of this rhomboid is to prevent the accumulation of membrane aggregates of substrate(s) especially when they are non-functional. Therefore, we performed cellular fractionation to characterise the location of HybA and established whether it forms aggregates following its over-expression in cells as an orphan protein (*i.e.* in the absence of HybB). Successful fractionation was determined by analysing preparations for the presence of TolA (for the whole-cell and membrane fractions) and RpoB (for the

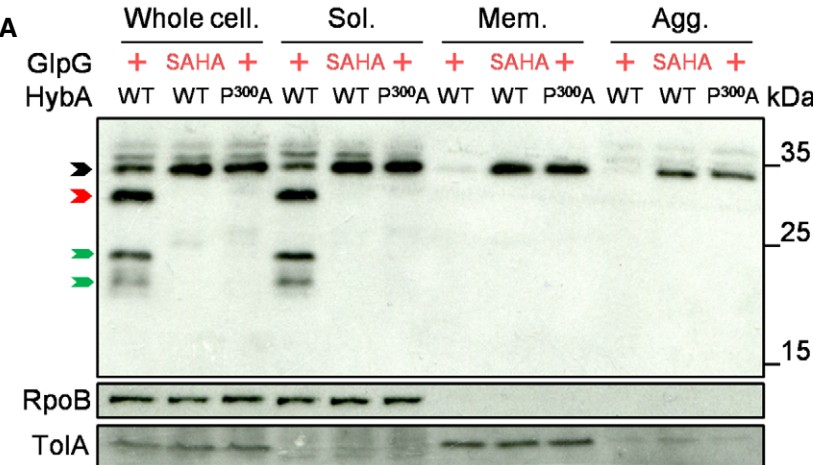

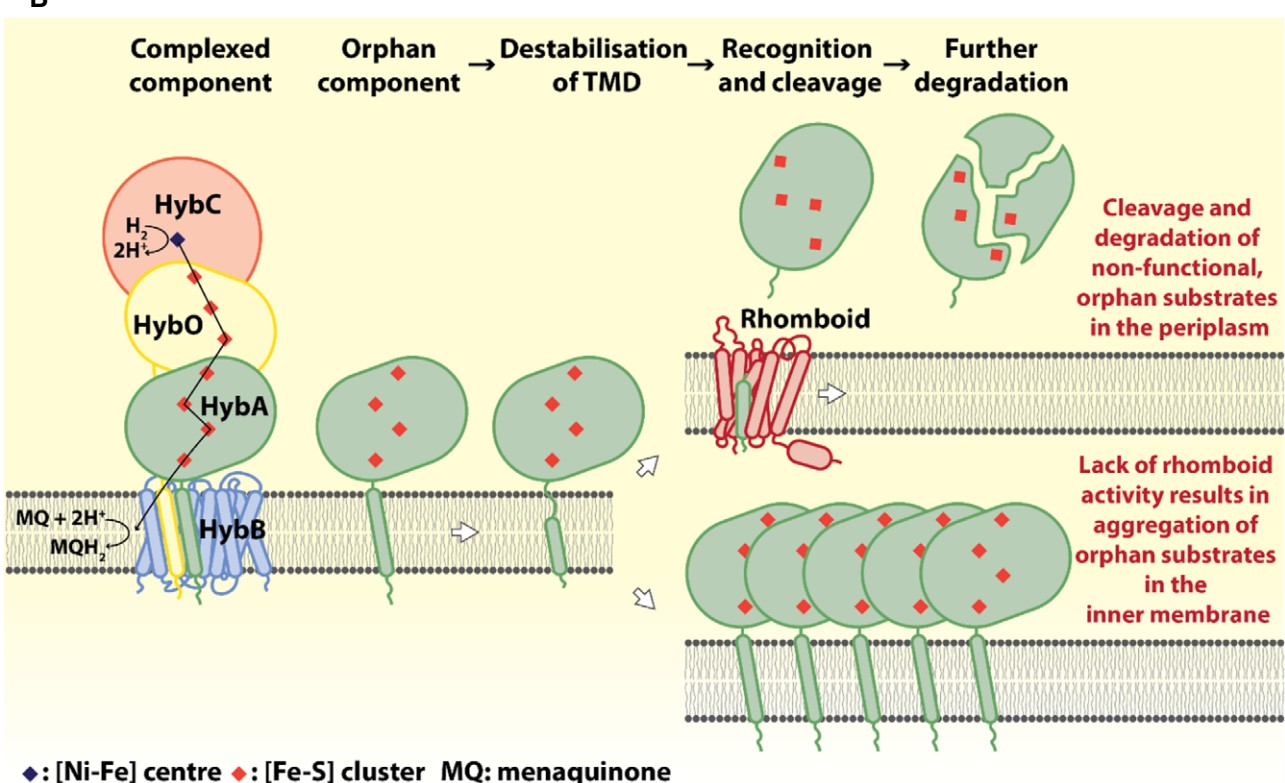

**Figure 7. Rhomboids prevent aggregation of orphan substrates in the inner membrane.**

A Western blot analysis probing the localisation and status of plasmid-encoded N-terminally V5-tagged wild-type (WT) or modified (P[300]A) HybA in *S. sonnei* Δ*rhom7*Δ*hybB* with wild-type (+) or inactive (SAHA) GlpG expressed from pUC19. Whole-cell lysate (Whole cell.), Soluble (Sol.), Membrane (detergent-solubilised, Mem.) and the Aggregate (Agg.) fractions are shown. HybA that is uncleaved, cleaved only by GlpG and further degraded post-GlpG cleavage is marked by black, red and green arrows, respectively.

B Model of rhomboid-mediated quality control by selectively targeting orphan components of multiprotein respiratory complexes.

Source data are available online for this figure.

whole-cell and soluble fractions) by Western blot analysis (Fig 7A) (Berardini *et al*, 1999; Le Maire *et al*, 2000; Garibyan, 2003; Rassam *et al*, 2018). Of note, while no aggregates were detected in the

presence of GlpG, we found that in the absence of GlpG, orphan HybA was retained in membranes where it formed aggregates (Fig 7A). Consistent with GlpG enabling the elimination of orphan

substrates from the membrane, significant amounts of HybA[P300A] (which cannot be cleaved by GlpG) also became aggregated in the membranes of cells even in the presence of active GlpG (Fig 7A), indicating that GlpG enables the elimination of orphan substrates from membranes, preventing their aggregation.

## Discussion

In this study, we characterised two rhomboids in *S. sonnei*, GlpG and Rhom7, which share sequence homology with *E. coli* GlpG and *P. stuartii* AarA, respectively. We found that both GlpG and Rhom7 are active enzymes that selectively target orphan components of respiratory complexes. The exquisite specificity of both rhomboids for orphan proteins allows them to participate in membrane quality control, by initiating the degradation and elimination of substrates only when they are not part of a functional complex. The search for rhomboid substrates in bacteria has been challenging. Our study highlights potential reasons for this. The substrates are only expressed during growth in specific conditions (*e.g.* in low oxygen environment for HybA and FdnH), cleavage can be prevented by the presence of partner proteins, and because GlpG and Rhom7 selectively target non-functional proteins, their absence might not lead to robust phenotypes.

Rhomboids typically contain a core domain consisting of six TMDs, with or without a 7[th] TMD, and either an additional N-terminal or C-terminal domain (Koonin *et al*, 2003; Stevenson *et al*, 2007; Saita *et al*, 2017). The role of the 7[th] TMD in rhomboids has not been investigated previously. We demonstrate that the 7[th] TMD of Rhom7 and its C-terminal domain are dispensable for cleavage of an artificial substrate. The 7[th] TMD could localise the C-terminal domain to the periplasm, or be involved in substrate recognition or regulate proteolytic activity (see accompanying manuscript). It has been proposed that rhomboids are among the most rapidly diffusing molecules within lipid bilayers (Kreutzberger *et al*, 2019), enabling them to patrol membranes in search of substrates. Consistent with this, GlpG is enzymatically inefficient (with a $k_{cat} \approx 0.006/s$) due to its catalytic dyad, rather than the triads found at active sites of classical serine proteases (Dickey *et al*, 2013). This low efficiency would limit indiscriminate cleavage of membrane proteins (Dickey *et al*, 2013), consistent with our findings that GlpG and Rhom7 are quality control enzymes which specifically target orphan membrane proteins. A key requirement of successful quality control is that it should not perturb a functional system, supported by GlpG having no effect on Hyd-2 activity under normal growth conditions (Fig 4F and G), even though HybA is an essential component of this complex.

Our initial bioinformatic screen for identifying substrates excluded substrates predicted to have multiple TMDs, while almost half of our 44 candidates were predicted to be components of multiprotein complexes. The initial screen using TMHMM identified FdnH; however, reiterative searches with HHpred were needed to find FdoH and HybA (Fig 2A), highlighting potential limitations of bioinformatic searches relying on a single algorithm. We screened the TMDs from all candidates by over-expressing them in an AS, so they were likely to be orphan proteins. Despite this, TMDs from only six of the candidates were cleaved by GlpG and/or Rhom7, highlighting the specificity of these rhomboids. We identified HybA and FdoH as the first physiological substrates for the highly characterised enzyme GlpG

(Wang *et al*, 2006; Moin & Urban, 2012; Zoll *et al*, 2014; Cho *et al*, 2016), while HybA and FdnH are substrates for Rhom7 only when this protease is expressed from a plasmid as endogenous expression levels of Rhom7 are insufficient to result in detectable cleavage. The three rhomboid substrates we identified share several characteristics: (i) they are all components of multiprotein membrane complexes (Abaibou *et al*, 1995; Jormakka *et al*, 2002; Dubini & Sargent, 2003); (ii) they have an N-terminal periplasmic ferredoxin domain containing four [4Fe-4S] clusters; and (iii) they are secreted as folded proteins by the Tat system (Sargent *et al*, 2002). Interestingly, the only other known substrate of a bacterial rhomboid is TatA, an integral part of the Tat secretion system (Stevenson *et al*, 2007). This suggests that there might be an evolutionary link between bacterial rhomboids and the Tat system, with the possibility that GlpG and Rhom7 cleave incorrectly folded proteins which have been secreted by Tat, and thence fail to associate into a membrane complex, making them susceptible to rhomboid cleavage.

Our work reveals a form of quality control, which is based on four distinct steps (Fig 7B): (i) a substrate becomes an orphan membrane protein through either non-incorporation into or dissociation from its partner protein(s); (ii) destabilisation of the orphan substrate TMD; (iii) recognition and cleavage by a rhomboid; and finally (iv) further cleavage of the substrate in the periplasm. In this way, bacterial rhomboids offer an elegant strategy to monitor the functional status of components of membrane complexes and protect cells from any potential danger posed by orphan proteins through their accumulation, mis-incorporation into non-cognate complexes and/or localisation to inappropriate cellular sites (Juszkiewicz & Hegde, 2018). We found that in the absence of GlpG, orphan HybA accumulates in membranes as aggregates (Fig 7A). Of note, TTC19 is a substrate of the mitochondrial rhomboid PARL (Saita *et al*, 2017; Spinazzi *et al*, 2019) and is important for the function and assembly of mitochondrial complex III (cIII) by regulating the turnover of the [2Fe-2S] containing Rieske protein (Bottani *et al*, 2017), a crucial catalytic component of cIII. Together with our work, this suggests that the link between rhomboids and quality control of respiratory complexes may be conserved throughout evolution.

Due to the irreversible nature of proteolysis, rhomboid activity in membranes must be carefully regulated. In eukaryotes, this can be achieved by compartmentalisation of enzymes and their substrates in different organelles; for example, rhomboid-1 is located in the endoplasmic reticulum and only cleaves Spitz that has been delivered to this site from the Golgi apparatus (Urban *et al*, 2001). Rhomboid activity can also be influenced by environmental cues such as cytosolic $Ca^{2+}$ concentrations (Baker & Urban, 2015). Our work illustrates another mechanism of controlling rhomboid activity through the selective cleavage of non-functional, orphan membrane proteins. This mechanism is reminiscent of regulation by compartmentalisation, but as bacteria lack organelles, compartmentalisation is achieved by sequestration of the substrate TMD in a protected molecular niche provided by its partner protein(s).

A similar mechanism is involved in the quality control of the αβ T-cell receptor (αβ-TCR) (Klausner *et al*, 1990). The TMD of the α-chain of the αβ-TCR harbours two positively charged residues (Lys and Arg), making it energetically unfavourable in the membrane. However, the α-chain is stabilised by the presence of Asp residues in the TMDs of other components of the αβ-TCR (Feige and Hendershot, 2013). Although the αβ-TCR is evolutionarily and functionally

distinct from bacterial respiratory complexes, the quality control of both systems relies on partner proteins stabilising TMDs which contain atypical features for a lipophilic environment, such as charged residues in the αβ-TCR and prolines in HybA, FdoH and FdnH. Interestingly, *E. coli* GlpG can cleave truncated but not full-length MdfA, a multidrug transporter with 12 TMDs, with cleavage occurring at a type I TMD of MdfA possessing a proline (Erez & Bibi, 2009); this suggests that other rhomboids might also monitor and target metastable TMDs in multipass membrane proteins (see accompanying manuscript).

Although poorly understood, assembly-dependent quality control of bacterial protein complexes in the cytoplasmic membrane also involves regulated proteolysis. In *E. coli* and related organisms, this is mainly mediated by the AAA$^+$ protease, FtsH (Bittner *et al*, 2017). FtsH has been implicated in degradation of membrane proteins that lack periplasmic domains, such as SecY, PspC and $F_o$a, when they are dissociated from their native complexes (Kihara *et al*, 1995; Akiyama *et al*, 1996; Singh & Darwin, 2011). However, FtsH does not degrade inner membrane proteins with tightly folded periplasmic domains (Kihara *et al*, 1999). Of note, all rhomboid substrates we identified possess a folded periplasmic ferredoxin domain (Abaibou *et al*, 1995; Jormakka *et al*, 2002; Beaton *et al*, 2018) and so are likely to be refractory to FtsH-mediated quality control. Rhomboids could therefore have evolved to cleave orphan substrates with periplasmic domains, complementing the activity of FtsH or indeed working in concert with other proteases as illustrated in the accompanying manuscript.

Hyd-2 and formate dehydrogenase N are required for the virulence of several enteric pathogens (Maier *et al*, 2013; Winter *et al*, 2013). Of note, HybA, FdnH and FdoH all possess 4Fe-4S clusters, which are prone to damage by exogenous stress. Therefore, the contribution of GlpG and Rhom7 to quality control and membrane protein homeostasis might be enhanced when substrates are subject to insults such as host innate immune response or metal stress (Slauch, 2011; Goldblatt, 2014; Dong *et al*, 2015). Indeed, for FdnH and FdoH, we only observed further degradation following exposure to copper stress (Fig 6E and F). Therefore, it may be that the *in vivo* colonisation defect exhibited by *E. coli* lacking GlpG results from its failure to deal with accumulated components of respiratory complexes in the membrane (Russell *et al*, 2017). Furthermore, efforts to determine the mechanisms of rhomboid catalysis have been hampered by the lack of rhomboid substrate pairs for enzymatic and structural studies. Our work identifying two physiological substrates of GlpG should enable studies of the scission process (Cho *et al*, 2016) and may facilitate the search for more substrates of bacterial rhomboids. Finally, given their efficiency in an evolutionarily ancient organism, the ability of bacterial rhomboids to recognise and selectively cleave orphan membrane proteins might mean that similar mechanisms act to regulate critical events within membranes in higher organisms.

# Materials and Methods

## Media, bacterial strains and plasmids

Bacterial strains, plasmids and primers used in this study are listed in Tables EV2–EV4, respectively. *Escherichia coli* DH5α was used for cloning and *E. coli* SM10 λ*pir* (Simon *et al*, 1983) for maintenance of plasmids for conjugation. Escherichia coli was grown at 37°C with agitation at 180 r.p.m. in LB media or on LB agar plates containing 1.5% bacteriological agar (w/v). Antibiotics were used at the following concentrations: carbenicillin, 100 μg/ml; streptomycin, 50 μg/ml; and chloramphenicol, 20 μg/ml.

For aerobic growth, S. sonnei CS14 (Holt *et al*, 2012) and derivatives were grown at 37°C with agitation at 180 r.p.m in LB media. Anaerobic growth was performed in a Whitley A35 Anaerobic Station with 5% $H_2$, 10% $CO_2$, 85% $N_2$ or 10% $H_2$, 10% $CO_2$, 80% $N_2$ at 37°C with agitation at 180 r.p.m. in relevant media. Viable bacteria (colony-forming units/ml) were enumerated by serially diluting cultures in 96-well plates. An aliquot of 10 μl from each well was plated onto LB agar and incubated at 37°C overnight before counting.

## Construction of mutants

To construct markerless mutants, we generated vectors for each mutation using pCONJ4s which was constructed with NEBuilder HiFi DNA Assembly Cloning Kit (New England Biolabs) as follows. The backbone of pKNG101 (Kaniga *et al*, 1991) was amplified with primers pGL225/pGL224 and assembled with the *bla* gene amplified from pGEM-T Easy (Promega) using pGL222/pGL223, generating pCONJ1. pCONJ1 was linearised by *Spe*I digestion (New England Biolabs) and a *Pme*I restriction site added with primers pGL185/pGL186, yielding pCONJ2. Subsequently, the backbone of pCONJ2 was amplified with pGL154/pGL97 and assembled with *sfGFP* (Pedelacq *et al*, 2006) amplified with primers pGL98/pGL155 from pNCC1 sfGFP (Wormann *et al*, 2016), generating pCONJ3. pCONJ3 was then linearised by *Eco*RV digestion (New England Biolabs) and assembled with *sacB* amplified from pKNG101 using pGL247/pGL248, giving rise to pCONJ4. Finally, pCONJ4 was digested by *Afl*II (New England Biolabs) and assembled with a primer heterodimer formed by pGL342/pGL343, to form pCONJ4s. The upstream and downstream flanking regions (approximately 500 bp each) of each target gene were amplified by PCR with the relevant primers and assembled into pCONJ4s at the *Pme*I site; constructs were verified by sequencing (Source BioScience).

*Shigella sonnei* CS14 mutants were then generated by conjugation. First, 20 μl of stationary-phase *E. coli* SM10 λ*pir* harbouring a pCONJ4s derivative was mixed with 20 μl of stationary-phase *S. sonnei* recipient, and the entire volume was spotted onto LB agar. Bacteria were incubated at 21°C until the spots dried and then at 37°C for 4 h, after which the bacteria were scraped from the plate and resuspended in 1 ml of LB, and 100 μl was plated onto LB agar plates with antibiotics. Plates were incubated at 37°C for 14 h. Next, a single colony was used to inoculate liquid LB with carbenicillin and streptomycin and grown to stationary phase. Bacteria were collected by centrifugation, washed and diluted fivefold into 5 ml LB, and then grown for 4 h at 37°C. 100 μl of 50-fold diluted culture was then plated on salt-free LB agar containing 10% (w/v) sucrose to select for loss of integrated plasmid; colonies were analysed by PCR to identify mutants and confirmed by sequencing (Source BioScience).

pBAD33-based vectors (Guzman *et al*, 1995) were constructed by assembling appropriate *glpG* and *rhom7* fragments into *Xba*I/ *Sph*I-linearised pBAD33 by NEBuilder (New England Biolabs).

Specific point mutations or addition of a sequence encoding a 3xHA tag was introduced by Gibson cloning (New England Biolabs) with appropriate primers. pUC19-based vectors (Yanisch-Perron *et al*, 1985) were constructed by assembling appropriate *glpG* fragments into *Sph*I/*Hind*III-linearised pUC19 by NEBuilder, and Gibson cloning was used to introduce specific point mutations or sequences encoding a triple-HA tag. pLAC101 is a low-copy-number plasmid for expressing proteins under IPTG induction. pLAC101 contains the pSC101 origin (amplified with pGL1315/pGL1316 from pUA139) (Zaslaver *et al*, 2006), the kanamycin resistance gene and *lacI* with a cloning site amplified from pNCC1 (Wormann *et al*, 2016) with primers pGL897/pGL1314 and pGL894/pGL1314, respectively. pLAC101V5-*hybA*/V5-*hybA*$_{P300A}$ were generated by assembling appropriate fragments into *PacI*/*Kpn*I-linearised pLAC101. Constructs were verified by sequencing (Source BioScience). Vectors were introduced into *S. sonnei* by electroporation using a Pulse Controller Plus (Bio-Rad) (800 Ωs, 25 µF, 2.5 kV) in 2-mm-gap electroporation cuvettes (Molecular BioProducts) and transformants selected on LB agar plates containing appropriate antibiotics and 0.2% (w/v) L-arabinose (Sigma) where relevant.

## Rhomboid-dependent protein cleavage assays

Substrate cleavage assays were performed in wild-type *S. sonnei*, *S. sonnei* lacking *glpG* and/or *rhom7*, with strains expressing the rhomboids with or without protein tags. In plasmid-based assays, rhomboids were expressed from pBAD33 with arabinose induction, while substrates were expressed using pKS508. For assays performed under aerobic conditions, 50 µl of overnight culture was added to 5 ml LB, and bacteria were grown at 37°C, 180 r.p.m. for 75 min before addition of L-arabinose at a final concentration of 0.2% (w/v). Bacteria were then grown for a further 2 h and harvested for SDS–PAGE and Western blotting. Assays performed under anaerobic conditions were performed in the same way except that 100 µl of overnight culture was added to 5 ml of LB supplemented with 0.5% (w/v) fumarate and 0.5% (v/v) glycerol, and bacteria were grown for 3 h before harvesting for Western blot analysis. L-arabinose (0.2% w/v) was added 1 h after subculturing. Copper stress was achieved by adding CuCl$_2$ (400 µM final concentration) for 30 min prior to harvesting bacteria. Protein translation inhibition was achieved by adding chloramphenicol (final concentration, 100 ng/ml) to exponential-phase cultures.

## HybA localisation and aggregation status

Assays were performed with *S. sonnei* Δ*rhom7*Δ*hybB* containing N-terminally V5-tagged wild-type (WT) or modified (P$^{300}$A) HybA expressed from pLAC101 with IPTG induction together with wild-type (+) or inactive (SAHA) GlpG constitutively expressed from pUC19. One ml of bacteria grown overnight anaerobically was added to 50 ml LB supplemented with 0.5% (w/v) fumarate and 0.5% (v/v) glycerol, and bacteria were grown anaerobically for 2 h before the addition of IPTG (1 mM). Bacteria were incubated for another hour and then pelleted. Bacterial pellets were resuspended and incubated in 2 ml lysis buffer (50 mM Tris–HCl, 200 mM NaCl, 1 mM EDTA, 2 mM phenylmethylsulfonyl fluoride, pH 7.4) supplemented with lysozyme (1 mg/ml), DNase I (0.05 mg/ml) and protease inhibitor cocktail (Roche) for 30 min at room temperature.

Bacteria were sonicated on ice and spun at 1,000× *g* for 30 min at 4°C. A 30 µl aliquot of supernatant was collected as the "whole-cell" fraction for Western blot analysis, while the remaining 970 µl was ultra-centrifuged at 45,000 *x g* for 30 min. After ultra-centrifugation, 30 µl of the supernatant was collected as the "Soluble" fraction while the rest was decanted. The pellet was resolubilised by 970 µl lysis buffer supplemented with 1% Triton X-100 (v/v) at 4°C for 30 min prior to another round of ultra-centrifugation at 45,000× *g* for 30 min (Le Maire *et al*, 2000). After that, 30 µl of the supernatant was collected as the "Membrane" fraction. The pellet was resuspended and boiled in 940 µl SDS–PAGE sample buffer as the "Aggregates" fraction for Western blotting analysis.

## SDS–PAGE and Western blotting

Whole-cell lysates were prepared by boiling bacteria for 10 min in SDS–PAGE sample buffer (50 mM Tris, 0.75% (w/v) SDS, 10% (v/v) glycerol, 25 mM EDTA, 1% (v/v) β-mercaptoethanol, 0.11 mM bromophenol blue) at a dilution of an OD$_{600}$ unit per 333 µl. A 20 µl aliquot of lysate was separated by electrophoresis (SDS–PAGE) on 12% polyacrylamide gels at 150 V for 75 min in SDS–PAGE buffer (25 mM Tris, 192 mM glycine, 0.1% (w/v) SDS, pH 8.5). Proteins were transferred to nitrocellulose membranes using the Trans-Blot Turbo System (Bio-Rad). Tagged proteins were detected using anti-FLAG M2 antibody (Sigma, F3165, 1:1,000) and goat anti-mouse IgG-HRP (Bio-Rad, 172-1011, 1:5,000), or an anti-V5 (D3H8Q) antibody (Cell Signaling Technology, #13202, 1:1,000) and goat anti-rabbit IgG-HRP (Santa Cruz Biotechnology, sc-2004, 1:5,000). RecA was detected using anti-RecA antibody (Abcam, ab63797, 1:5,000) and goat anti-rabbit IgG-HRP (Santa Cruz Biotechnology, sc-2004, 1:5,000). Bands were detected by chemiluminescence (ECL, GE Healthcare), and band intensity was quantified using Photoshop CC (Lynda.com) as follows: a section of the blots including the relevant protein bands was generated and the total number of pixels within each box was measured, which correlates with the amount of protein.

## Bioinformatic identification of potential substrates

Type I and type III topology proteins were identified using the TMHMM Server v. 2.0 (Krogh *et al*, 2001). As TMDs of proteins are structurally similar (*e.g.* they contain non-polar residues favouring formation of alpha-helices), we used HHpred (Soding *et al*, 2005) to identify structural homologues of the 16 predicted type I and type III proteins identified by initial analysis with TMHMM. Amino acid sequences of the TMD of the 16 candidates were screened against the *E. coli* K-12 proteome (07_Mar version, the HHpred server does not include a *Shigella* proteome), using the PDB_mmCIF70_3_Jul database and default parameters (Soding *et al*, 2005). Orthologues (> 90% sequence identity) of proteins that are also present in *S. sonnei* were analysed by Phobius 1.01 (Kall *et al*, 2007) and TOPCONS 2.0 (Tsirigos *et al*, 2015) to exclude proteins with multiple TMDs.

## Construction and screen of the TMD library of potential rhomboid substrates

pKS508 was linearised by digestion with *Kpn*I and *Xba*I and ligated with annealed primers encoding TMDs + 16 additional amino acids of 44 predicted type I and III membrane proteins. Plasmids encoding

the TMDs are listed in Table EV3 and were verified by sequencing (Source BioScience) before being co-transformed with pBAD33 encoding active/inactive GlpG/Rhom7 into *S. sonnei* Δ*glpG*Δ*rhom7*. Cleavage of the 44 artificial substrates by GlpG or Rhom7 *in vivo* was assessed once. Assays were repeated with all artificial substrates that displayed more product/less substrate after expression of an active rhomboid compared to an inactive enzyme.

## Cleavage site identification

Wild-type *S. sonnei*, or *S. sonnei* in which the chromosomal copy of *glpG* was edited to encode the inactive mutant S201A (S201A), both expressing HybA fused to the sfCherry-3xFLAG tag at the C-terminus, was grown anaerobically to an $OD_{600} = 0.5$, collected by centrifugation and lysed by CellLytic B Cell Lysis Reagent (Sigma) supplemented with lysozyme (Sigma) and protease inhibitor cocktail (Roche). The chimeric HybA substrate and its cleavage products were isolated by affinity chromatography using ANTI-FLAG M2 Affinity Agarose Gel (Sigma) and eluted into 50 mM Tris, 150 mM NaCl, 15% glycerol. Purified proteins were separated on SDS–PAGE, electroblotted onto a PVDF PSQ membrane (Millipore) and subjected to sequential Edman degradation on a Procise Protein Sequencing System (491 Protein Sequencer, PE Applied Biosystems).

## Bacterial two-hybrid assay

Bacterial two-hybrid assay was performed based on a previously developed system with CyaA, an adenylate cyclase from *Bordetella pertussis* with modifications (Karimova *et al*, 1998). T25 and T18 fragments of *B. pertussis* CyaA were fused to the C-terminus of HybA and HybB and expressed from their native site on the chromosome of *S. sonnei* lacking endogenous CyaA. A 20 µl aliquot of stationary-phase bacteria was spotted on LB agar supplemented with 0.5% (w/v) fumarate, 0.5% (v/v) glycerol, 1 mM IPTG and 40 µg/ml X-gal. Spots were dried and incubated either aerobically or anaerobically with 10% $H_2$, 10% $CO_2$, 80% $N_2$ at 37°C for 14 h.

## Assays of hydrogenase-2 activity

*Shigella sonnei* possesses three potential hydrogenases, Hyd-1 (encoded by *hyaA-E*), Hyd-2 and Hyd-3 (encoded by the *hyc* operon) (McNorton & Maier, 2012; Pinske & Sawers, 2016). *Shigella sonnei* Δ*hyaA-F*Δ*hycE*Δ*rhom7* was generated to assess the impact of GlpG and HybA cleavage on Hyd-2 activity. Growth of strains was measured under anaerobic conditions with $H_2$ and fumarate as previously described (Dubini *et al*, 2002). One millilitre of overnight cultures of *S. sonnei* grown aerobically in LB at 37°C with agitation at 180 r.p.m. was centrifuged at 21,100× *g* for 1 min, the bacterial pellets were washed twice with M9 minimal media supplemented with 0.5% fumarate, 12.5 µg/ml nicotinic acid and 0.2% casamino acids, and resuspended in the same media, and the number of bacteria was quantified by measuring the $OD_{600}$. Cultures were transferred to a Whitley A35 Anaerobic Workstation supplied with 5% $H_2$, 10% $CO_2$ and 85% $N_2$ and diluted to a starting $OD_{600}$ of 0.01 in 50 ml of supplemented minimal media that had been pre-incubated in 5% $H_2$, 10% $CO_2$ and 85% $N_2$ overnight. Bacteria were grown at 37°C with agitation at 180 r.p.m. and the $OD_{600}$ measured hourly for 24 h.

Hydrogen uptake was measured by gas chromatography. *Shigella sonnei* strains were prepared as described above and then diluted in 20 ml of supplemented minimal media to an $OD_{600} = 0.01$ in glass gas chromatography vials in a Whitley A35 Anaerobic Workstation. Vials were sealed with #25 Suba-Seal Septa (Sigma-Aldrich) and sealed with Parafilm before removal from the anaerobic chamber. The headspace was purged with 10% $H_2$ and 90% argon (BOC gases) for 10 min (100 scc/min total flow rate) through a mass flow controller (Sierra Instruments) using a Sterican neural therapy needle (0.80 × 120 mm, 21 G × 4¾'', B Braun) at room temperature before transfer to a shaking incubator at 37°C, 250 r.p.m. Samples (15 µl) of the headspace gas were taken periodically using a Hamilton syringe, and the amount and composition of the sampled gas were analysed using a Gas Chromatograph System 7890A (Agilent Technologies) with ShinCarbon ST 100/120 Column, and $N_2$ as the carrier gas, at 32°C. Each run was stopped approximately 3 min post-injection to ensure capture of the entire $H_2$ peak. Prior to measurements, a 15-µl air injection was applied to verify the gas chromatograph. At each time point, two independent samples of headspace were analysed for each test sample and one sample was analysed for the negative control. After each injection, the Suba-Seal was reinforced with Parafilm. The amount of $H_2$ in the headspace was calculated by establishing a standard curve with known concentrations of $H_2$, and measuring the area underneath the $H_2$ peak in experiments, quantified by Agilent EZChrom Elite. Uptake of $H_2$ at a given time point $T_x$ is calculated by the formula: $H_2$ uptake $T_x$ = amount of $H_2$ in the headspace at $T_0$ - amount of $H_2$ in the headspace at $T_x$.

## Analysis of Type Three Secretion System activity

Overnight cultures of *S. sonnei* grown aerobically in LB were used to inoculate 10 ml TSB and grown at 37°C with agitation at 180 r.p.m until an $OD_{600}$ of 1.5 was reached. Bacteria were harvested by centrifugation at 21,100× *g*, washed and resuspended in PBS to an $OD_{600}$ of 5. Congo red was added (200 µg/ml final concentration) to the bacterial suspension, which was incubated at 37°C for 15 min to induce secretion. Bacteria were then incubated on ice for 5 min, centrifuged at 21,100× *g* and 150 µl of supernatant extracted, mixed with an equal volume of 2× SDS–PAGE sample buffer and heated at 100°C for 10 min prior to electrophoresis (20 µl) on 10% acrylamide gels. Proteins were stained using SilverXpress™ Silver Staining Kit (Thermo Fisher Scientific).

## Oxidative stress assays

For $H_2O_2$ stress, 500 µl of an overnight culture of *S. sonnei* grown aerobically in LB was used to inoculate 50 ml LB and grown until the $OD_{600}$ reached 0.5, then 5 ml was aliquoted into 50 ml falcon tubes. Bacteria were pre-incubated with 200 µM $H_2O_2$ for 20 min before the addition of $H_2O_2$ (final concentration 2 mM) and incubated for a further 15 min before enumeration of viable cells by plating to LB agar.

For paraquat (PQ), 500 µl of overnight cultures of *S. sonnei* grown aerobically in LB was used to inoculate 50 ml M9 minimal media (supplemented with 0.4% glucose, 12.5 µg/ml nicotinic acid) and grown to an $OD_{600}$ of 0.5; then, 5 ml was aliquoted into 50-ml tubes. Bacteria were then incubated with 200 µM PQ for 15 min

before the addition of PQ to a final concentration of 10 mM and incubated for another 45 min before enumeration by plating.

## Statistical analysis

Two-way ANOVA was used to calculate *P* values for $OD_{600}$ measurements. One-way ANOVA was used to calculate *P* values for CFU counts and Western blot band intensities. Both one-way and two-way ANOVA values were calculated by GraphPad Prism.

**Expanded View** for this article is available online.

## Acknowledgements

We thank Professor Frank Sargent for helpful discussions, Dr Daniel Mesquita Da Fonseca for helpful discussions and experimental guidance, and Dr Joanna Szczepaniak from the for TolA anti-serum. GL was funded by an EPA Scholarship; work in CMT's laboratory is supported by the Wellcome Trust and BBSRC. Rhiannon Evans is supported by BBSRC grant (BB/N006321/1) awarded to FAA. SEB is grateful to the Dakota Foundation. KS acknowledges support from the ERDF/ESF project No. CZ.02.1.01/0.0/0.0/16_019/0000729, from Czech Science Foundation (project no. 18-09556S) and from the Czech Academy of Sciences (RVO: 61388963), and thanks Petra Rampírová and Zdeněk Voburka for technical assistance.

## Author contributions

GL, SEB, RME, CMT conceived and designed experiments. GL performed most experiments and analysed data with help from MR; both were supervised by RME. SEB and RE performed hydrogen uptake/generation experiments with GL; KS performed N-terminal sequencing on cleaved HybA. AGG, MF, RE, KS and FAA provided suggestions and contributed to experimental design. GL, RME, and CMT wrote the paper with input from all co-authors.

## Conflict of interest

The authors declare that they have no conflict of interest.

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
