## [Review Process File · The EMBO Journal]

Bacterial rhomboid proteases mediate quality control of orphan membrane proteins

Guangyu Liu, Stephen Beaton, Adam Grieve, Rhiannon Evans, Miranda Rogers, Kvido Strisovsky, Fraser Armstrong, Matthew Freeman, Rachel Exley and Christoph Tang

Review timeline:

Submission date:	12 th July 2019
Editorial Decision:	29 th August 2019
Revision received:	22 nd January 2020
Editorial Decision:	19 th February 2020
Revision received:	17 th March 2020
Accepted:	24 th March 2020

Editor: Hartmut Vodermaier

Transaction Report:

1st Editorial Decision

29th August 2019

Thank you again for submitting your manuscript on bacterial rhomboid-mediated orphan membrane protein degradation to The EMBO Journal. With some delay linked to the summer vacation period and the necessity of finding referees ready to take on two back-to-back papers, we have now received the comments of three expert reviewers. As you will see, all referees consider your findings potentially interesting and the key conclusions important in principle. At the same time, the reports bring up several major concerns that in our view currently preclude publication in The EMBO Journal. The two main issues are the unclear physiological importance of the proposed GlpG/Rhom7 quality control function, noted by both referees 1 and 2, and the missing evidence for rhomboid cleavage being strictly required for subsequent degradation of orphan subunits, as detailed by referee 1.

Given the interest of the subject, we concluded that we would like to give you the opportunity to address these key points, as well as the various more specific issues brought up by the reviewers, by way of a revised version of the manuscript. I should however stress that it will be essential to obtain more definitive support for the rhomboid involvement in orphan protein degradation, as well as to get at least some indications of a physiological role of this mechanism, e.g. by looking for cellular and/or molecular phenotypes under particular perturbed conditions. I realize that this may require considerable further time and effort, and would therefore be happy to discuss a possible extension of the default three-months revision deadline, during which publication of any competing/related work would as usual not have a negative impact on our final assessment of your own study. Please be reminded that our policy to allow only a single round of major revision will make it important to comprehensively answer to all points raised at this stage. I would further encourage you to contact me already during the early stages of revision to discuss any proposals for addressing the reviewers' concerns

 REFEREE REPORTS

Referee #1:

This study presents a search and analysis of substrates for the rhomboid proteases GlpG and Rhom7 in *Shigella sonnei*. Using a starting list of 44 TMDs from single-pass membrane proteins in the appropriate topology, the authors use an over-expression strategy to find those that are cleaved by GlpG and/or Rhom7 when the TMD and flanking region is placed into a reporter cassette. This approach identified 6 proteins, of which the authors primarily pursue further studies on HybA and to a lesser extent, FdoH. After demonstrating that endogenous HybA is a target for GlpG, the authors provide evidence that only the population of HybA that is not in a complex with HybB is cleaved. A similar result is seen for FdoH and FdnH. The authors therefore conclude that GlpG and Rhom7 can cleave 'orphans' of multiprotein complexes. It has long been known that orphans are degraded in the absence of their binding partners, and this study implicates rhomboid family members as being involved in this process for certain types of substrates. This conclusion, if supported, is a notable advance suitable for publication in EMBO J.

At this stage however, there are two major issues that diminish my support. First, as far as I can tell, there is no strong evidence that GlpG/Rhom7-mediated cleavage of orphaned HybA, FdoH, or FdnH is an obligate step in their degradation or that their degradation is appreciably impaired in the absence of the rhomboids. This is a central conclusion of the study and it is essential that the authors address this convincingly and completely before publication. Second, no phenotypic consequence is documented under any condition when these rhomboids are removed. It seems to me that with three substrates in hand and new insight into the function of GlpG/Rhom7, the authors should be able to document at least some consequence of their absence. For example, over-expression of the orphans in the absence of their binding partner should lead to aggregation or membrane stress or sensitivity to protein misfolding stress preferentially when their degradation pathway is eliminated (i.e., strains lacking GlpG/Rhom7). In the absence of any consequence and without a clear documentation that there is indeed a degradation defect (my first point), one is left wondering what the biological significance is of the observation that GlpG/Rhom7 cleaves orphaned HybA, FdoH, and FdnH.

Major comments:

1) The authors' central claim that GlpG 'licences' HybA degradation is not rigorously established. It is important to demonstrate that HybA, in the absence of HybB, does not get degraded effectively in the absence of GlpG. The current data leaves open the possibility that HybA is degraded equally well regardless of whether it is cleaved by GlpG or not. The experiment in Fig. 4D with the mutant GlpG is really not convincing for two reasons. First, to my eye, it actually seems like HybA is being lost similarly in the GlpG mutant cells as it is in the GlpG wild type cells (hard to tell as the blot is over-exposed to judge the mutant lanes). Second, the mutant protein might protect HybA from degradation relative to the situation in the absence of GlpG. For this reason, it seems important to directly test the idea that GlpG is required for orphan HybA degradation. Similarly, FdoH or FdnH cleavage should not automatically be equated with an obligate step in its degradation. I agree that the authors have shown that orphan FdoH and FdnH are cleaved, but they have not demonstrated that in the absence of cleavage, degradation of FdoH or FdnH orphans is actually impaired. This needs to be documented in order to draw the central conclusion in this study.

2) Somewhat related to point 1, it is important to document some type of phenotypic consequence of preventing GlpG/Rhom7 cleavage. The authors have fluorescent protein tagged versions of their orphans and strains lacking their binding partners. So at the least, one should be able to see if the proteins accumulate and aggregate in the absence of their cleavage, and whether this has some proteostasis type phenotype.

Minor points:

1) The characterisation of the screen as "genome-wide" is misleading. It is in fact a candidate screen of 44 pre-selected candidates based on making several assumptions including: (i) that substrates would be single-pass proteins; (ii) that substrates would be recognised and cleaved when their TMD regions are analysed out of context; (iii) that their topology predictions about proteins that were excluded from analysis are correct. The authors should therefore avoid the term 'genome-wide' and be explicit about the assumptions they have made.

- 2) How many of the 38 proteins that were not identified as substrates part of multi-protein complexes? In other words, are the 6 substrates selectively enriched in protein complex subunits? This should be discussed in the text.
- 3) The authors state in Fig. 3B that there is GlpG/Rhom7 independent cleavage of HybO. How do they know the lower bands are a consequence of cleavage? Perhaps better to just re-state as "no evidence for GlpG/Rhom7-dependent cleavage was observed" then say in the figure legend that the identity of the lower bands is not known.
- 4) Fig. 6C might be labelled incorrectly. What is the difference between lanes 1&2 versus 3&4? Similarly, what is the difference between lanes 5&6 versus 7&8? They cannot simply be replicates because the result is different between lane 5 vs. 7, yet they are labelled identically. Please clarify.
- 5) I personally do not feel "complex protection" is a good term. The Rhomboids are not protecting anything and being within a complex isn't the only way proteins can be protected from Rhomboids. Perhaps something like "orphan licencing" is preferable, although I really don't see why one needs to coin a new term for this well-established phenomenon of orphan degradation.
- 6) In Fig. 4B, when HybA is over-expressed, there seems to be an increase in total cleavage, but not in the proportion of HybA that is cleaved. Shouldn't the proportion and the total amount increase? This is not evident in the gel that is shown. Perhaps GlpG is saturated? Please clarify.
- 7) Fig. 4G is somewhat confusing and is perhaps labelled incorrectly. Why are cleavage products seen in lanes 5-8? According to the labels, these have mutant GlpG, yet cleavage and degradation are higher than in lanes 1-4. Please clarify.
- 8) Fig 5 documents that no phenotype is seen when HybA cleavage is manipulated by GlpG expression levels or by mutating HybA's cleavage site. The authors should clarify in the text how sensitive the growth and H₂ uptake assays are. In other words, would a 50% difference in Hyd-2 complex levels show a phenotype? Would a 10% difference be detected? I ask because if the growth assay is not very sensitive, then the observations are not especially meaningful since GlpG might be affecting Hyd-2 fairly substantially and still not be detected.

Referee #2:

The paper by Guangyu Liu and co-workers uses an exhaustive but elegant genetic approach to screen and identify physiological substrates for the GlpG and Rhom7 homologues from *Shigella sonnei*.

The authors identified GlpG and Rhom7 by sequence homology and confined their ability to cleave an artificial substrate by expressing these proteins from a plasmid in a strain lacking chromosomal copies of *glpG* and *rhom7*. As a control they could show that plasmids expressing catalytically inactive mutants failed to cleave the model substrate. They next screened for a phenotype for the dual *glpG/rhom7* deletion, but were unable to detect any impact under a variety of conditions.

Unperturbed, they manually screened for likely substrates based on the known specificity. TMDs from these likely substrates were re-screened in their model system (replacing the model TM of TatA). They could detect protease cleavage from two protein belonging to the Hyd-2 complex. To validate these potential substrates, they made strains in which chromosomal HybO or HybA were tagged with a sfCherry-3xFLAG tag to allow detection of cleavage. In this setup, HybA showed some cleavage (about 20%) by GlpG/Rhom7. They further reasoned that HybA cleavage might be protected as it forms a complex with HybB, and indeed they were able to show efficient cleavage if *hybB* gene was deleted or HybA was over expressed (and therefore had uneven stoichiometry). As a control they they should that under physiological conditions wherein the hydrogenase is switched on, the GlpG protease did not cleave functional HybA. Based on the confirmed relationship between GlpG protease activity for the perturbed complexes, they could show that subunits of formate dehydrogenase O were also sensitive to proteolysis by the rhomboids when they were "missing" their partners. They conclude that GlpG/Rhom7 rhomboids mediate quality control by aiding the

degradation of misassembled respiratory complexes.

Overall, despite the lack of a clear phenotype, the work highlights novel substrates for rhomboid proteases. Although interesting, I think they may have "over-claimed" their final model from the work presented.

Major comments

While the work is excellent, there is still no clear physiological function of GlpG/Rhom7 since the degradation of substrates observed is only under "artificial" non-physiological conditions. Since there is no evidence for mis-regulated expression of multi complex proteins, I do not think the authors can justify using the term "complex protection".

Minor comments:

1. Are these substrates really TAT dependent? Since they have a single TM then should they to be Sec-TAT dependent ... at least this was my understanding from Tracey Palmers group (eLife. 2017; 6: e26577.). Perhaps, it is therefore the unusual biogenesis combination that is sensitive to quality control by the rhomboid proteases and not those inserted strictly Sec or TAT-dependent proteins.
2. The "UXKGUUXP motif" seems generic enough to be non-consequential. What is the scientific basis for this motif?

Referee #3:

Despite detailed structural insights, very little is known about the cognate substrates and biological functions of rhomboid proteases in bacteria. Two co-submitted manuscripts now fill this gap, one identifying substrates of two rhomboids in the Gram-negative pathogen *Shigella sonnei* and one revealing a rhomboid function in the Gram-positive model bacterium *Bacillus subtilis*. Both manuscripts present exciting results and are valuable contributions to the field. Since they report strikingly different substrates (orphan single-pass transmembrane proteins versus a multi-pass membrane transporter) and biological functions, the discussions would benefit from cross-referencing and comparing the results in case both papers appear back-to-back.

Liu et al. report that orphan membranes proteins (that are not incorporated in functional complexes) are recognized and eliminated by rhomboids. Assuming that rhomboid substrates are single-pass membrane proteins, the authors used a bio-computational screen to identify putative substrates. They narrowed down a list of 44 candidates to three components of membrane respiratory complexes that were cleaved when dissociated from their interaction partners but not touched when correctly assembled. The authors term this mechanism "complex protection". As indicated above this is an excellent manuscript. It presents a substantial amount of work, is well written and easy to follow. I only have a few minor questions and comments.

1. Some results and statements seem inconsistent, e.g. HybO looks like a true substrate in Fig. 2 (inactive rhomboids reduce cleavage) but is no substrate in Fig. 3 (identical cleavage in the absence or presence of rhombids). Is it due to different conditions: aerobic versus anaerobic? Page 5, line 3 states that HybA and FdoH are GlgG substrates and FdnH is a Rhom7 substrate. Later it is shown that HybA is a substrate of both GlgG and Rhom7 (page 9, line 9). Matters are complicated by the fact that endogenous levels of Rhom7 are insufficient for detectable cleavage activity (page 13). Please amend the text carefully. In addition, it might help to label the true substrates in Fig. 2E, e.g. by bold letters.
2. Page 9, line 3 from the bottom: The authors state that GlgG virtually cleaves all HybA when it is an orphan protein. While it is true that the full-length protein disappears (Fig. 4D), there is no increase of the cleavage product at all. Please explain.
3. Are the lanes in Fig. 6G mis-labeled? Shouldn't the processed band appear in lane 5 but not in lane 3?

4. *P. stuartii* TatA appears to be the best rhomboid substrate (Fig. 2). How does its TMD sequence compare to the ones of FdoH, FdnH and HybA (Fig. 7)?

5. The bioinformatic search for rhomboid proteases proceeded in two steps. The first step resulted in 16 candidates, the second in 28 additional candidates. I was wondering whether the three confirmed substrates derived from the initial step but was unable to find that information.

1st Revision - authors' response

22nd January 2020

Referee #1:

This study presents a search and analysis of substrates for the rhomboid proteases GlpG and Rhom7 in Shigella sonnei. Using a starting list of 44 TMDs from single-pass membrane proteins in the appropriate topology, the authors use an over-expression strategy to find those that are cleaved by GlpG and/or Rhom7 when the TMD and flanking region is placed into a reporter cassette. This approach identified 6 proteins, of which the authors primarily pursue further studies on HybA and to a lesser extent, FdoH. After demonstrating that endogenous HybA is a target for GlpG, the authors provide evidence that only the population of HybA that is not in a complex with HybB is cleaved. A similar result is seen for FdoH and FdnH. The authors therefore conclude that GlpG and Rhom7 can cleave 'orphans' of multiprotein complexes. It has long been known that orphans are degraded in the absence of their binding partners, and this study implicates rhomboid family members as being involved in this process for certain types of substrates. This conclusion, if supported, is a notable advance suitable for publication in EMBO J.

We are grateful to the reviewer for their comments.

At this stage however, there are two major issues that diminish my support. First, as far as I can tell, there is no strong evidence that GlpG/Rhom7-mediated cleavage of orphaned HybA, FdoH, or FdnH is an obligate step in their degradation or that their degradation is appreciably impaired in the absence of the rhomboids. This is a central conclusion of the study and it is essential that the authors address this convincingly and completely before publication.

In response to the reviewer's comment, we now demonstrate the contribution of rhomboids to the degradation of orphaned HybA, FdoH, and FdnH as suggested. We introduced an N terminal tag onto each of the substrates to analyse and assess further cleavage of their periplasmic domains following initial rhomboid cleavage with or without their membrane partners. We show that degradation of orphan HybA is dependent on cleavage by GlpG (Fig. 6A and B). For FdoH and FdnH, we only observed processing of beyond initial rhomboid cleavage when bacteria were exposed to copper stress which is known to perturb the Fe-S clusters in these enzymes (Fig 6E and F for FdoH and FdnH, respectively) (Macomber & Imlay, 2008). Similar to HybA, further degradation of these substrates is dependent on their initial processing by rhomboids (lines 255-259, 260-266).

Second, no phenotypic consequence is documented under any condition when these rhomboids are removed. It seems to me that with three substrates in hand and new insight into the function of GlpG/Rhom7, the authors should be able to document at least some consequence of their absence. For example, over-expression of the orphans in the absence of their binding partner should lead to aggregation or membrane stress or sensitivity to protein misfolding stress preferentially when their degradation pathway is eliminated (i.e., strains lacking GlpG/Rhom7). In the absence of any consequence and without a clear documentation that there is indeed a degradation defect (my first point), one is left wondering what the biological significance is of the observation that GlpG/Rhom7 cleaves orphaned HybA, FdoH, and FdnH.

As suggested by the reviewer, we examined whether the rhomboids prevent the aggregation of orphan substrates. To address this issue, we fractionated cells following overexpression of orphan HybA with or without active GlpG; we also examined the fate of uncleavable HybA. The extent of aggregation was assessed by failure of proteins to be solubilised by 1% Triton X-100 (Le Maire, Champeil et al., 2000). In the absence of GlpG, we found that orphan HybA accumulates into aggregates in the membranes while the presence of active GlpG, HybA did not form aggregates (Fig 7A); uncleavable HybA also also aggregates even when GlpG is present (lines 268-281). We also include further comments about the possible role of GlpG based on the known colonisation defect of an *E. coli glpG* mutant (lines 389-392) (Russell, Richards et al., 2017).

Major comments:

1) The authors' central claim that GlpG 'licences' HybA degradation is not rigorously established. It is important to demonstrate that HybA, in the absence of HybB, does not get degraded effectively in the absence of GlpG. The current data leaves open the possibility that HybA is degraded equally well regardless of whether it is cleaved by GlpG or not. The experiment in Fig. 4D with the mutant GlpG is really not convincing for two reasons. First, to my eye, it actually seems like HybA is being lost similarly in the GlpG mutant cells as it is in the GlpG wild type cells (hard to tell as the blot is over-exposed to judge the mutant lanes).

We agree that the intensity of the band corresponding to uncleaved HybA does appear to decrease over time in bacteria lacking GlpG (Fig 4D). However, the amount of uncleaved HybA is actually stable in this strain when HybA levels are normalised using the signal from RecA, the loading control. We agree with the reviewer that this is an important point, so we now also show this quantification in a new Figure (Fig EV7B) that confirms that processing is dependent on cleavage by GlpG.

Second, the mutant protein might protect HybA from degradation relative to the situation in the absence of GlpG. For this reason, it seems important to directly test the idea that GlpG is required for orphan HybA degradation. Similarly, FdoH or FdnH cleavage should not automatically be equated with an obligate step in its degradation. I agree that the authors have shown that orphan FdoH and FdnH are cleaved, but they have not demonstrated that in the absence of cleavage, degradation of FdoH or FdnH orphans is actually impaired. This needs to be documented in order to draw the central conclusion in this study.

Please see our responses to the reviewer's general comments (see above). We have now analysed the degradation of all our substrates in the presence/absence of the rhomboid responsible for their initial cleavage. We include these findings in Fig 6A, B, E and F and discuss our results in lines 255-259, 260-266, of our revised manuscript.

2) Somewhat related to point 1, it is important to document some type of phenotypic consequence of preventing GlpG/Rhom7 cleavage. The authors have fluorescent protein tagged versions of their orphans and strains lacking their binding partners. So at the least, one should be able to see if the proteins accumulate and aggregate in the absence of their cleavage, and whether this has some proteostasis type phenotype.

We have addressed this point in our responses to the reviewer's general comments. We now show that the absence of GlpG leads to the accumulation of orphan HybA in the membrane fraction of cells. We also observed accumulation of orphan uncleavable HybA in the presence of GlpG. These data are included in Fig 7A (lines 268-281).

Minor points:

1) The characterisation of the screen as "genome-wide" is misleading. It is in fact a candidate screen of 44 pre-selected candidates based on making several assumptions including: (i) that substrates would be single-pass proteins; (ii) that substrates would be recognised and cleaved when their TMD regions are analysed out of context; (iii) that their topology predictions about proteins that were excluded from analysis are correct. The authors should therefore avoid the term 'genome-wide' and be explicit about the assumptions they have made.

We agree with the reviewer's comment and have deleted 'genome-wide' from our description of the screen, and now include the assumptions we made for our screen in our revised manuscript (line 311-315).

2) How many of the 38 proteins that were not identified as substrates part of multi-protein complexes? In other words, are the 6 substrates selectively enriched in protein complex subunits? This should be discussed in the text.

We thank the reviewer for raising this interesting point. Out of the six substrates identified from our initial screen, only HybA, FdoH, and FdnH were shown to be genuine rhomboid substrates and they are all part of multi-protein respiratory

complexes (Abaibou, Pommier et al., 1995, Jormakka, Törnroth et al., 2002, Pinske, Jaroschinsky et al., 2015). Of the other 38 potential substrates from our screen, 15 are predicted to be components of multicomponent complexes. As our initial substrate screen employed over-expression of the TMDs only within an artificial substrate, the TMDs were likely to be orphans. Despite this, these TMDs were not cleaved by GlpG or Rhom7, highlighting the specificity of these rhomboids. We discuss this point in our revised article (lines 315-318).

3) *The authors state in Fig. 3B that there is GlpG/Rhom7 independent cleavage of HybO. How do they know the lower bands are a consequence of cleavage? Perhaps better to just re-state as "no evidence for GlpG/Rhom7-dependent cleavage was observed" then say in the figure legend that the identity of the lower bands is not known.*

We have altered the text as suggested by the reviewer (lines 183-187).

4) *Fig. 6C might be labelled incorrectly. What is the difference between lanes 1&2 versus 3&4? Similarly, what is the difference between lanes 5&6 versus 7&8? They cannot simply be replicates because the result is different between lane 5 vs. 7, yet they are labelled identically. Please clarify.*

We apologise for this error and have relabelled our figures and changed the legend (now Fig 5C).

5) *I personally do not feel "complex protection" is a good term. The Rhomboids are not protecting anything and being within a complex isn't the only way proteins can be protected from Rhomboids. Perhaps something like "orphan licencing" is preferable, although I really don't see why one needs to coin a new term for this well-established phenomenon of orphan degradation.*

In line with the reviewer's comments, we have removed the term 'complex protection' from our manuscript, and instead describe rhomboids as orphan targeting enzymes.

6) *In Fig. 4B, when HybA is over-expressed, there seems to be an increase in total cleavage, but not in the proportion of HybA that is cleaved. Shouldn't the proportion and the total amount increase? This is not evident in the gel that is shown. Perhaps GlpG is saturated? Please clarify.*

We have quantified the ratio of cleaved to uncleaved HybA following over-expression of HybA. The proportion of cleaved to uncleaved HybA increases by around 3-fold compared to when HybA is not over-expressed. Therefore, disrupting HybA/HybB stoichiometry by overexpressing HybA leads to increased GlpG cleavage of HybA (Figure EV7A, lines 208-210).

7) *Fig. 4G is somewhat confusing and is perhaps labelled incorrectly. Why are cleavage products seen in lanes*

5-8? According to the labels, these have mutant GlpG, yet cleavage and degradation are higher than in lanes 1-4. Please clarify.

Thank you. We have relabelled our figures and changed the legend (now Figure 6B) as suggested.

8) Fig 5 documents that no phenotype is seen when HybA cleavage is manipulated by GlpG expression levels or by mutating HybA's cleavage site. The authors should clarify in the text how sensitive the growth and H₂ uptake assays are. In other words, would a 50% difference in Hyd-2 complex levels show a phenotype? Would a 10% difference be detected? I ask because if the growth assay is not very sensitive, then the observations are not especially meaningful since GlpG might be affecting Hyd-2 fairly substantially and still not be detected.

We appreciate the reviewer's concerns and have changed the wording to reflect that the growth assays are less sensitive assays (Dubini, Pye, et al. 2002, Pinske, Jaroschinsky et al., 2015) while the hydrogen uptake assays are more sensitive and should be directly proportional to the number of active Hyd-2 enzymes that are connected to the quinone pool via HybA (lines 222-223).

Referee #2:

The paper by Guangyu Liu and co-workers uses an exhaustive but elegant genetic approach to screen and identify physiological substrates for the GlpG and Rhom7 homologues from Shigella sonnei.

The authors identified GlpG and Rhom7 by sequence homology and confined their ability to cleave an artificial substrate by expressing these proteins from a plasmid in a strain lacking chromosomal copies of glpG and rhom7. As a control they could show that plasmids expressing catalytically inactive mutants failed to cleave the model substrate. They next screened for a phenotype for the dual glpG/rhom7 deletion, but were unable to detect any impact under a variety of conditions.

Unperturbed, they manually screened for likely substrates based on the known specificity. TMDs from these likely substrates were re-screened in their model system (replacing the model TM of TatA). They could detect protease cleavage from two protein belonging to the Hyd-2 complex. To validate these potential substrates, they made strains in which chromosomal HybO or HybA were tagged with a sfCherry-3xFLAG tag to allow detection of cleavage. In this setup, HybA showed some cleavage (about 20%) by GlpG/Rhom7. They further reasoned that HybA cleavage might be protected as it forms a complex with HybB, and indeed they were able to show efficient cleavage if hybB gene was deleted or HybA was over expressed (and therefore had uneven stoichiometry). As a control they they should that under physiological conditions wherein the hydrogenase is switched on, the GlpG protease did not cleave functional HybA. Based on the confirmed relationship between GlpG protease activity for the perturbed complexes, they could show that subunits of formate dehydrogenase O were also sensitive to proteolysis by the rhomboids when they were "missing" their partners. They conclude that GlpG/Rhom7 rhomboids mediate quality control by aiding the degradation of misassembled respiratory complexes.

Overall, despite the lack of a clear phenotype, the work highlights novel substrates for rhomboid proteases. Although interesting, I think they may have "over-claimed" their final model from the work presented.

Major comments

While the work is excellent, there is still no clear physiological function of GlpG/Rhom7 since the degradation of substrates observed is only under "artificial" non-physiological conditions. Since there is no evidence for mis-regulated expression of multi complex proteins, I do not think the authors can justify using the term "complex protection".

Please see our responses to reviewer 1: i) we have deleted the term ‘complex protection’ from our revised manuscript. Instead, we state that rhomboids selectively target orphan membrane proteins; and ii) we show that the rhomboids initiate removal of non-functional orphan proteins from membranes (Fig 6B, E and F), and failure to do so leads to aggregation of the substrates in the membranes (Fig 7A). However this was not associated with a reduction in bacterial survival (not shown).

Minor comments:

1. Are these substrates really TAT dependent? Since they have a single TM then should they to be Sec-TAT dependent ... at least this was my understanding from Tracey Palmers group (eLife. 2017; 6: e26577.). Perhaps, it is therefore the unusual biogenesis combination that is sensitive to quality control by the rhomboid proteases and not those inserted strictly Sec or TAT-dependent proteins.

We thank the reviewer for raising this interesting point. According to Tooke *et al* (Tooke, Babot *et al.*, 2017), integral membrane Tat substrates fall into two categories - those that are N-terminally anchored in the bilayer by a non-cleaved signal sequence, such as the Rieske Fe-S protein of *Paracoccus*, and those bearing a single C-terminal TMD such as HybA, FdnH, and FdoH (Hatzixanthis, Palmer *et al.*, 2003).

FdoH and FdnH do not contain a Tat signal peptide, but are transported to the periplasm in complex with Tat-transported FdoG and FdnG, respectively. Tooke *et al.* specifically demonstrated the coordinated membrane insertion of Sec-Tat dual-dependent substrates from proteins belonging to the ‘Rieske protein’ category (*i.e.* proteins with at least one Sec-dependent TMD preceding the Tat signal peptide, and a C-terminal Tat-cargo domain) (Tooke *et al.*, 2017). However, all our substrates fall into the other category. Therefore while the Sec system might be involved in the membrane insertion of our substrates, there is no conclusive evidence for this. Therefore, we refer to our substrates as ‘Tat-dependent’.

2. The "UXKGUUXP motif" seems generic enough to be non-consequential. What is the scientific basis for this motif?

In line with the reviewer's comment, we have re-drawn the figure and removed the text describing the 'UXKGUUXP motif' in the manuscript.

Referee #3:

*Despite detailed structural insights, very little is known about the cognate substrates and biological functions of rhomboid proteases in bacteria. Two co-submitted manuscripts now fill this gap, one identifying substrates of two rhomboids in the Gram-negative pathogen *Shigella sonnei* and one revealing a rhomboid function in the Gram-positive model bacterium *Bacillus subtilis*. Both manuscripts present exciting results and are valuable contributions to the field. Since they report strikingly different substrates (orphan single-pass transmembrane proteins versus a multi-pass membrane transporter) and biological functions, the discussions would benefit from cross-referencing and comparing the results in case both papers appear back-to-back.*

Liu et al. report that orphan membranes proteins (that are not incorporated in functional complexes) are recognized and eliminated by rhomboids. Assuming that rhomboid substrates are single-pass membrane proteins, the authors used a bio-computational screen to identify putative substrates. They narrowed down a list of 44 candidates to three components of membrane respiratory complexes that were cleaved when dissociated from their interaction partners but not touched when correctly assembled. The authors term this mechanism "complex protection".

As indicated above this is an excellent manuscript. It presents a substantial amount of work, is well written and easy to follow. I only have a few minor questions and comments.

We are grateful to the reviewer for their comments.

1. *Some results and statements seem inconsistent, e.g. HybO looks like a true substrate in Fig. 2 (inactive rhomboids reduce cleavage) but is no substrate in Fig. 3 (identical cleavage in the absence or presence of rhombids). Is it due to different conditions: aerobic versus anaerobic?*

We apologise for any confusion when comparing results in Fig. 2 and Fig. 3. Fig. 2 shows results of cleavage of only the TMD of HybO in an artificial substrate (Fig. 2B), while Fig. 3 shows the cleavage of a tagged version of full length HybO. When analysing HybO cleavage by GlpG, we detected a 36 kDa band in the absence of GlpG (Fig. 3B). As this is the predicted size of the product generated by GlpG cleavage of HybO, we could not assess whether full length HybO is a rhomboid substrate. We have altered our revised article to clarify this point (lines 178-180).

Page 5, line 3 states that HybA and FdoH are GlgG substrates and FdnH is a Rhom7 substrate. Later it is shown that HybA is a substrate of both GlgG and Rhom7 (page 9, line 9). Matters are complicated by the fact

that endogenous levels of Rhom7 are insufficient for detectable cleavage activity (page 13). Please amend the text carefully. In addition, it might help to label the true substrates in Fig. 2E, e.g. by bold letters.

We thank the reviewer for their comments and have altered the text as suggested (lines 100, 170, and 173-174). We have not changed the labelling of figures, as the ‘true substrates’ are identified later in the paper and not at this point.

2. Page 9, line 3 from the bottom: The authors state that GlgG virtually cleaves all HybA when it is an orphan protein. While it is true that the full-length protein disappears (Fig. 4D), there is no increase of the cleavage product at all. Please explain.

We agree that there is no increase in the cleavage product in Fig 4D. This is because there is further degradation of the cleavage product; please see our comments to reviewers 1 and 2 and data in Fig. 6A and B together with changes to the text (lines 254-257).

3. Are the lanes in Fig. 6G mis-labeled? Shouldn't the processed band appear in lane 5 but not in lane 3?

We thank the reviewer for bringing this to our attention. The reviewer is correct, and we have altered the figure as suggested (now Figure 5G).

4. P. stuartii TatA appears to be the best rhomboid substrate (Fig. 2). How does its TMD sequence compare to the ones of FdoH, FdnH and HybA (Fig. 7)?

We thank the reviewer for their comments, and have included the alignment of the TMD from *P. stuartii* TatA in Fig 5D together with the substrates we identified. We highlight the presence of a proline residue in the TMDs from all these rhomboid substrates (lines 241-243).

5. The bioinformatic search for rhomboid proteases proceeded in two steps. The first step resulted in 16 candidates, the second in 28 additional candidates. I was wondering whether the three confirmed substrates derived from the initial step but was unable to find that information.

This is an interesting point. FdnH was identified as one of the first 16 substrates by TMHMM, while HybA and FdoH were picked up in re-iterative searches due to their homology with FdnH. We describe this in our revised manuscript (lines 313-315).

References

- Abaibou H, Pommier J, Benoit S, Giordano G, Mandrand-Berthelot M-A (1995) Expression and characterization of the *Escherichia coli fdo* locus and a possible physiological role for aerobic formate dehydrogenase. *Journal of bacteriology* 177: 7141-7149
- Hatzixanthis K, Palmer T, Sargent F (2003) A subset of bacterial inner membrane proteins integrated by the twin-arginine translocase. *Molecular Microbiology* 49: 1377-1390
- Jormakka M, Törnroth S, Byrne B, Iwata S (2002) Molecular Basis of Proton Motive Force Generation Structure of Formate Dehydrogenase-N. *Science* 295: 1863-1868
- Le Maire M, Champeil P, Møller JV (2000) Interaction of membrane proteins and lipids with solubilizing detergents. *Biochimica et Biophysica Acta - Biomembranes* 1508: 86-111
- Macomber L, Imlay JA (2008) The iron-sulfur clusters of dehydratases are primary intracellular targets of copper toxicity. *Proceedings of the National Academy of Sciences of the United States of America* 106: 8344-8349
- Pinske C, Jaroschinsky M, Linek S, Kelly CL, Sargent F, Sawers RG (2015) Physiology and bioenergetics of [NiFe]-hydrogenase 2-catalyzed H₂-consuming and H₂-producing reactions in *Escherichia coli*. *J Bacteriol* 197: 296-306
- Russell CW, Richards AC, Chang AS, Mulvey MA (2017) The Rhomboid Protease GlpG Promotes the Persistence of Extraintestinal Pathogenic *Escherichia coli* within the Gut. *Infect Immun* 85
- Tooke FJ, Babot M, Chandra G, Buchanan G, Palmer T (2017) A unifying mechanism for the biogenesis of membrane proteins co-operatively integrated by the Sec and Tat pathways. *Elife* 6

2nd Editorial Decision

19th February 2020

Thank you for submitting your revised manuscript on orphan membrane protein QC by bacterial rhomboid proteases. All three original referees have now assessed the new version and your responses once more (see below), and I am pleased to say have no further scientific concerns. Following addressing of several formal/editorial issues, we shall therefore be happy to accept the manuscript for EMBO Journal publication.

REFeree REPORTS

Referee #1:

The authors have addressed my concerns.

Referee #2:

Overall, I think the study does enough to convince that the mis-folded components of respiratory complexes (HybA and FdnH) are GlpG substrates. Whether these are "physiological" substrates I think will take further studies. Nevertheless, this paper pushes the field in a new direction to test this.

Referee #3:

The revised version presents new experiments and the authors responded adequately to all comments.

2nd Revision - authors' response

17th March 2020

The Authors have made the requested editorial changes.

Accepted

24th March 2020

Thank you for submitting your final revised manuscript for our consideration. I am pleased to inform you that we have now accepted it for publication in The EMBO Journal.

Corresponding Author Name: Prof. Christoph Tang, Dr. Rachel M Exley

Journal Submitted to: The EMBO Journal

Manuscript Number: EMBOJ-2019-102922R1